



# Impact of agricultural management on soil aggregates and associated organic carbon fractions: Analysis of long-term experiments in Europe

Ioanna S. Panagea[1], Antonios Apostolakis[2], Antonio Berti[3], Jenny Bussell[4], Pavel Čermak[5], Jan Diels[1], Annemie Elsen[6], Helena Kusa[5], Ilaria Piccoli[3], Jean Poesen[1,7], Chris Stoate[4], Mia Tits[6], Zoltan Toth[8], and Guido Wyseure[1]

[1]Department of Earth and Environmental Sciences, KU Leuven, Leuven, 3001, Belgium
[2]Max-Planck-Institute for Biogeochemistry; Jena, 07745, Germany
[3]DAFNAE Department, University of Padova, Legnaro PD, 35020, Italy
[4]Game & Wildlife Conservation Trust, Allerton Project, Loddington, Leicester, LE7 9XE, UK
[5]Crop Research Institute, Prague 6-Ruzyne, 161 06, Czech Republic
[6]Soil Service of Belgium (BDB), Heverlee, 3001, Belgium
[7]Institute of Earth and Environmental Sciences, UMCS, Lublin, 20-718, Poland
[8]Institute of Agronomy, Georgikon Campus, Keszthely, Hungarian University of Agriculture and Life Sciences, Gödöllő, 2100,Hungary

**Correspondence:** Ioanna S. Panagea (ioanna.panagea@kuleuven.be)

**Abstract.** Inversion tillage is a commonly applied soil cultivation practice in Europe, which though has been blamed for deteriorating topsoil stability and organic carbon (OC) content. In this study, the potential to reverse these negative effects in the topsoil by alternative agricultural management practices are evaluated in five long-term experiments (running from 8 to 54 years the moment of sampling) in Europe. Topsoil samples (0 –15 cm) were collected and analysed to evaluate the effects

of conservation tillage (reduced and no-tillage) or increased organic inputs of different origin (farmyard manure, compost, crop residues) combined with inversion tillage on topsoil stability, soil aggregates and within these, OC distribution. Effects from the treatments on the two main components of organic matter i.e., particulate (POM) and mineral associated (MAOM), were also evaluated. Reduced and no-tillage practices, as well as the additions of manure or compost, increased the aggregates mean weight diameter (MWD) and topsoil OC, as well as the OC corresponding to the different aggregate size fractions. The

incorporation of crop residues had a positive impact on the MWD but a less profound effect on OC content both on total OC and on OC associated with the different aggregates. A negative relationship between the mass and the OC content of the microaggregates (53 – 250 $\mu$m) was identified in all experiments. There was no effect on the mass of the macroaggregates and the occluded microaggregates (mM) within these, while the corresponding OC contents increased with less tillage and more organic inputs. Inversion tillage led to less POM within the mM, whereas the different organic inputs did not affect it. In all

experiments where the total POM increased, the total soil organic carbon (SOC) was also affected positively. We concluded that the negative effects of inversion tillage on topsoil can be mitigated by reducing the tillage intensity or by adding organic materials, optimally combined with non-inversion tillage methods.





## 1  Introduction

Soil tillage by mouldboard ploughing which inverts and mixes the soil is a classical conventional agricultural practice in tem-
perate agricultural systems, including most of Europe. This soil cultivation technique is used to prepare the soil for seeding, to
reduce soil compaction, to control weeds and crop residues and to improve water and nutrients availability for plants (Cannell,
1985; Peigné et al., 2018; Schneider et al., 2017; Hobson et al., 2021). An alternative common practice is disc ploughing which
breaks and mixes the soil but disturbs only the shallow soil horizons. Inverting the topsoil especially by mouldboard ploughing
is blamed for deteriorating the stability and arrangement of the soil aggregates (soil structure) with subsequent negative effects
on soil organic matter (SOM) preservation and sequestration potential (Six et al., 2000a; Six and Paustian, 2014). For this
reason, introducing agricultural practices that counter the adverse effects of mouldboard tillage on soil structure and SOM will
contribute to sustainable agricultural management and climate change mitigation (Cooper et al., 2021).

The protection of SOM in the soil aggregates has been proposed as an important stabilization mechanism. According to
Oades (1984), macroaggregates are initially formed around fresh plant material (i.e. particulate organic matter (POM)) enter-
ing the soil due to physical, chemical, and biological processes. As the organic matter within macroaggregates ($> 250$ $\mu$m)
decomposes, an internal structure develops consisting of microaggregates (53– 250 $\mu$m) within which the SOM is better pro-
tected (i.e., occluded SOM (mM-C)) as compared to non-occluded SOM (Ananyeva et al., 2013; De Clercq et al., 2015; Denef
et al., 2001; King et al., 2019). When macroaggregates break down, for instance due to wetting-drying cycles, the microaggre-
gates are released (Six et al., 2000a; Denef et al., 2001). However, agricultural practices such as inversion tillage may break
down soil macroaggregates prematurely, which results in the release of unstable microaggregates and leads to increased SOM
accessibility for decomposition (Kotronakis et al., 2017; Six et al., 2000a). For this reason, the fraction of microaggregates
held within the macroaggregates (mM) has been proposed as an indicator to monitor present and potential soil structural and SOM
changes caused by management practices (Denef et al., 2004, 2007; Kong et al., 2005; Six et al., 2000a; Six and Paustian,
2014). A parallel relationship in the change of the mass of the water-stable mM and the corresponding SOC (i.e., mM-C) has
been conceptualized (Six et al., 2000a) and identified (King et al., 2019; Modak et al., 2020; Zhao et al., 2018).

Tillage effects on soil structure and especially topsoil SOM are a point of controversy in the literature. Several studies ob-
served an increase in macroaggregates under non-inversion / zero till as compared to conventional inversion tillage agriculture
(Jat et al., 2019; Paustian et al., 1997; Six et al., 2000b). Reduced soil disturbance is often accompanied by an increase in SOC
in the topsoil (Apostolakis et al., 2017; Fuentes et al., 2012; Giannakis et al., 2014; Karlen et al., 2013; Virto et al., 2012) or
even the whole soil profile (Cooper et al., 2021; Varvel and Wilhelm, 2011) especially when it is combined with increased
inputs from cover crops or intercrops (Boddey et al., 2010; Fuentes et al., 2009; Minasny et al., 2017). This increase is mainly
explained by differences among the practices in terms of organic input accumulation in the topsoil (Andruschkewitsch et al.,
2014; Virto et al., 2012) and reduced mineralization rates. Other studies though, reported limited or no effects of reduced tillage
on SOM, especially when the entire soil profile was considered (Blanco-Canqui and Lal, 2008; Camarotto et al., 2020; Du et al.,
2017; Haddaway et al., 2017; Piccoli et al., 2016) which was explained mainly by the concentration of organic materials in the
top/surface layer due to no redistribution through inversion tillage in the deeper soil layers. Reduced and zero tillage have also





been claimed to cause other agricultural risks like increased soil compaction, impaired root development, increased weed and disease pressure and, possibly, lower crop yields and poor crop establishment (Berner et al., 2008; Ogle et al., 2012; Peigné et al., 2018; Piccoli et al., 2021; Van den Putte et al., 2010). Overall, while tillage reduction or cessation seems to alleviate the negative effects on soil structure and topsoil SOM, it also removes some of its beneficial effects especially in the short term. Kay and VandenBygaart (2002) and Sartori et al. (2021) speculated that in the short-term, soil compaction would be expected as a result of tillage absence and traffic load, while crop yield and soil structure stabilization are expected in the mid- and long-term as a result of greater biological activity and SOM redistribution.

While soil tillage increases the turnover rate of OM, a supply of organic materials should compensate for the losses. Under conventional agriculture, the addition of exogenous organic materials, like compost or manure, could offset the negative effects of inversion-tillage on soil structure (Williams et al., 2017) while maintaining the beneficial ones. Gross and Glaser (2021) concluded that regardless of tillage intensity manure application can increase the SOC stocks and that animal manure led to a greater SOC increase compared to green manure or plant-derived organic amendments (e.g., straw). While compost and animal manure additions are found to improve soil aggregation and increase both the total and the aggregate associated OC in croplands receiving conventional tillage, the inorganic fertilization did not result in such improved soil fertility (De Clercq et al., 2016; Kotronakis et al., 2017; Lin et al., 2019; Yin et al., 2016). Minasny et al. (2017) reviewed several studies and found that organic amendments (such as compost or manure) lead to a SOC accumulation of on average 0.5 Mg C ha$^{-1}$ year$^{-1}$ while residue incorporation leads to accumulation of on average 0.35 Mg C ha$^{-1}$ year$^{-1}$. Incorporation of crop residues increased total SOC from 2.7 up to 18.2 % (Bolinder et al., 2020) when compared to residue removal as well as increased the SOC within the different aggregate fractions (Zhao et al., 2018) and improved aggregation (Zhang et al., 2014). The effects of different residue management practices are highly connected with the tillage methods that are combined with it, thus leading to controversial findings when the latter is overlooked. Paul et al. (2013) concluded that tillage or residue management alone does not affect SOC, but when the residues are incorporated into the soil in the 15 – 30 cm layer, the SOC content increases whereas the aggregate stability is not improved. On the other hand, Li et al. (2020) identified non-inversion tillage (breaking the upper soil layers without any mixing or inverting) coupled with residue retention (when compared to conventional inversion or zero tillage with or without residue retention) as the optimum system to increase SOC stocks in the 0 – 30 cm soil layer.

Management effects on SOC and soil aggregation vary in the literature possibly due to differences in the climatic conditions and/or soil properties between studied regions (Pan et al., 2021), as well as, variations in the application and incorporation rates of the organic materials. Methodological differences between studies during sampling or analysis processes (Poeplau et al., 2018) may also be responsible for the controversial responses reported. For this reason, studies that investigate the impact of multiple and combined agricultural practices on several regions under a common methodological framework are needed. Such studies are in general scarce due to the logistical constraints associated with field experiments, and available studies in different countries are difficult to compare due to differences in objectives and analyses methods of heterogeneous field experiments.

SOM is a complex mixture that can be separated into multiple pools with distinct functional properties (Schrumpf et al., 2013; Trumbore, 2009). The separation of SOM into particulate organic matter (POM) that consists of plant-derived, relatively





undecomposed light fragments, and mineral-associated organic matter (MAOM) that consists of mostly simple molecules, has been proposed for many decades (Cambardella and Elliott, 1992; Poeplau et al., 2018). Lavallee et al. (2020) suggested the applicability of SOM separation into POM and MAOM for the investigation of management effects on carbon cycling. Coupling the separation of SOM in OM occluded in different aggregation fractions and in POM and MAOM within each aggregation fraction can help elucidating the effects of tillage intensity and soil amendments on SOM stability.

In this study, we investigated the effects of three agricultural practice categories on soil structure and SOC distribution in four aggregate size fractions and one subfraction (i.e. occluded microaggregates (mM)), for long-term experiments in five European countries following the same field and laboratory protocols. The agricultural practice categories were: i) soil cultivation including zero, minimum and conventional tillage, ii) crop residue incorporation or removal, iii) addition of exogenous organic or inorganic material including manure, compost and NPK fertilizers.

Our hypotheses were that in the topsoil:

- Inversion tillage reduces the mass of water-stable (large) macroaggregates and of SOC stabilized in macroaggregates and microaggregates, and this effect increases with increasing tillage intensity (from zero to conventional mouldboard ploughing).

- Matured exogenous organic fertilization, (i.e., manure and compost application), increases the mass of water-stable macroaggregates and increases SOC in all aggregate fractions, even under inversion tillage.

- Incorporation of the previous crop residues increases the mass of water-stable macroaggregates and occluded POM due to higher additions of fresh plant-derived organic matter, and thus leads to higher SOC stabilized in both occluded and free microaggregate, compared to the practice of crop residue removal.

- Increased organic inputs and lower soil disturbances lead to more water-stable mM and corresponding SOC (mM-C), (i.e., the mass of mM and occluded OC exhibit a linear relationship).

Our objectives were therefore: i) to evaluate the effects of the different treatments/practices on the topsoil structural stability and SOC, ii) to determine the soil mass and OC distribution in four water-stable aggregate size fractions (i.e., large macroaggregates: LM > 2000 $\mu$m; macroaggregates: 2000 <M < 250 $\mu$m; microaggregates: 250 <micro <53 $\mu$m; clay and silt domains: s&c <53 $\mu$m), iii) to determine the effects of the practices on the occluded microaggregates (mM) and the corresponding OC (mM-C) and iv) to determine the OC in POM and MAOM in the bulk soil and the aggregation fractions.

## 2 Methodology

### 2.1 Study site description

For our objectives, we took the option to combine several long-term experiments and to have a wide range of soil management practices and combinations rather than just concentrating on only one practice. Logistically, to handle seven experiments from



five countries by the same methods, and laboratory analysis by the same person we had to concentrate on the 0 – 15 cm
topsoil. Thus, topsoil samples (upper 15 cm) were collected from different long-term agricultural experiments (8 to 54 years)
with different treatments (the towns, countries, coordinates, main soil type and climate of the sites are given in Table A1
taken from Panagea et al. (2021)). The long-term experiments were set up independently from one another, with different
objectives and under different environmental conditions offering a wide range of representative management practices and
pedo-climatological conditions. For each country, the field experiment included management practices (and intensities) that
were adapted to the local conditions and were commonly applied by local farmers, but also intensities that were considered
extreme, to evaluate the maximum effects (e.g., 45 Mg ha$^{-1}$ compost annually in BE).

For this research, a subset of the experimental treatments was selected to include various treatment categories to test our
hypotheses (Table 1). This resulted in 79 experimental plots under 26 treatments. The first category includes primarily treat-
ments with different soil cultivation intensities (CZ, HU_2, UK), the second category focuses on the addition of different types
of exogenous organic materials (BE, IT_1c, IT_1p, HU_1,), in cropping systems that are conventionally tilled, and the third
category deals with the incorporation of crop residues (HU_1, IT2_c, IT_2l), also in systems under conventional inversion
tillage. The experiments in Italy are conducted on two different soil types each and, in this study, were analysed as separate
experiments: a clay and an initially peaty soil for IT_1 (i.e., IT_1c and IT_1p) and a clay and a loamy soil for experiment IT_2
(i.e., IT_2c and IT_2l). The selected treatments per experiment are presented in Table 1. At the five study sites, an identical
sampling and analysis procedure was performed for determining the soil aggregates size fraction distribution, the OC content
in each size fraction, and the separation of SOM into POM and MAOM (Figure 1) . For consistency reasons, all laboratory
analyses were conducted in the same laboratory with the same equipment, by the same person. This is particularly important
for the analysis of the aggregates.

## 2.2 Aggregate separation

Field-moist topsoil samples were taken with a sharp shovel in a Z-shape sampling design within each experimental plot from
the upper 15 cm of soil, mixed and directly broken to pass a <8 mm sieve. Soil samples were stored in plastic containers
to avoid compaction during transportation and then stored in the refrigerator until air-drying. All samples were air-dried and
stored in a dark and dry place at room temperature.

    Aggregate separation was done by wet sieving, as prescribed from Elliott (1986) and described in Six et al. (2000b). A
100 g subsample of 8 mm-sieved and air-dried soil was submerged in demineralized water for 5 min on a 2000 $\mu$m sieve.
Aggregates were separated by moving the sieve up and down mechanically 50 times in two minutes. The > 2000 $\mu$m fraction
was backwashed in a glass jar and the <2000 $\mu$m fraction was transferred in the next sieve (250 $\mu$m) and sieved with the same
methodology. This procedure was repeated with a 53 $\mu$m sieve. All aggregate fractions were oven-dried (50 °C) until dried,
weighted and stored for further fractionation and analysis. The aggregates were distinguished into large macroaggregates (>2
mm) (LM), macroaggregates (250 $\mu$m–2 mm) (M), microaggregates (53 $\mu$m–250 $\mu$m) (micro) and silt and clay size particles
(<53 $\mu$m) (s&c). The procedure was replicated three times for each sample from each experimental plot and the averaged
values used for the following analysis.



Table 1: Details of the treatments in the various experiments.

| Code | Treatments | Tillage system | Sampling month, year | Repl.(#), design | Main crop type |
|---|---|---|---|---|---|
| CZ | **Conventional:** Conventional ploughing (Turning of stubble—furrow opener at 10 cm, Mouldboard plough at 22 cm) (control) <br><br> **Minimum:** Minimum tillage (Turning of stubble- furrow opener at 10 cm, 30 % of crop residues remain on the soil surface <br><br> **Zero:** Zero tillage (all residues remain in the soil surface) | Depending on the experimental treatment | Nov 2018 | 4, † | Oil rapeseed, winter wheat, peas |
| HU_2 | **Conventional:** Deep winter ploughing (27–28 cm) + secondary tillage (control) <br><br> **Minimum:** Disking just before drilling (<15 cm) <br><br> **Shallow:** Shallow winter disking (<15 cm) +secondary tillage | Depending on the experimental treatment | Nov 2018 | 4, † | Winter wheat, maize |
| UK | **Conventional**: Ploughing at 25 cm (control) <br><br> **Direct drilling:** Direct drilling of the seeds into previous crop residues | Depending on the experimental treatment | Apr 2019 | 3, ‡ | Winter wheat, wheat, oat |
| BE | **No organic:** No organic fertilization (control) <br><br> **45tn3-yearly:** 45 Mg ha$^{-1}$ compost* applied every three years <br><br> **15tnyearly:** 15 Mg ha$^{-1}$ compost* applied yearly <br><br> **45tnyearly:** 45 Mg ha$^{-1}$ compost* applied yearly <br><br> *C/N ≈ 12 | Conventional tillage up to 23 cm and bed preparation according to the crop type | Oct 2019 | 4, ‡ | Winter wheat, carrots, sugar beet, potatoes |
| HU_1 | **NPK:** Only mineral fertilization/ removal of straw(control) <br><br> **NPK+FYM:** 35Mg ha$^{-1}$ 0.5% N, farmyard manure application every 3 years/removal of straw <br><br> **NPK+STR:** Straw and stalk incorporation completed with 10 kg N*Mg straw ha$^{-1}$ | Conventional tillage: shallow stubble tillage, ploughing (27 cm) in Autumn, secondary tillage and seedbed preparation | Nov 2018 | 3, † | Maize, winter wheat, winter barley |
| IT_1c | **Unfertilized:** No organic or mineral fertilization (control) <br> **Manure L1:** 20 Mg ha$^{-1}$ manure applied annually* <br><br> **Manure L2:** 40 Mg ha$^{-1}$ manure applied annually* | Shovelling-inversion tillage (0-20 cm) each autumn after the removal of crop residues. | Nov 2018 | 2, ‡ | Maize, winter wheat, potato, tillage radish (winter cover crop), ryegrass, silage maize |
| IT_1p | **Unfertilized:** No organic or mineral fertilization (control) <br> **Manure L1:** 20 Mg ha$^{-1}$ manure applied annually* <br><br> **Manure L2:** 40 Mg ha$^{-1}$ manure applied annually* <br><br> *Farmyard manure from dairy cows (20% dry matter, 0.5 % N, 0.25 % P$_2$O$_5$, 0.7 % K$_2$O) | | | | |



**Table 1 – continued from previous page**

| Code | Treatments | Tillage system | Sampling month, year | Repl.(#), design | Main crop type |
|------|-----------|----------------|---------------------|------------------|----------------|
| IT_2c | **Residue Remov.:** Removal of the previous crop residues (control) | Shovelling-inversion tillage (0-20 cm) each autumn | Nov 2018 | 3, ‡ | Maize, winter wheat, potato, ryegrass, silage maize, tillage radish |
|       | **Residue incorp.:** Burial of the previous crop residues | | | | |
| IT_2l | **Residue Remov.:** Removal of the previous crop residues (control) | | | | |
|       | **Residue incorp.:** Burial of the previous crop residues | | | | |

‡ Randomized complete block design (RCBD), † Split plot randomized complete block design (Split Plot-RCBD).

## 2.3 Determination of occluded microaggregates

The isolation of the microaggregates held within the macroaggregates was done by a microaggregate isolator using the method proposed by Six et al. (2000a). A 10 g subsample of macroaggregates was placed in the microaggregate isolator on top of a 250 $\mu$m mesh sieve, immersed in demineralized water and shaken with 50 glass beads (4 mm diameter). Steady and continuous water flow helped the microaggregates to flush on a 53 $\mu$m sieve. When only water was flushed in the sieve, the material on the 53 $\mu$m sieve was sieved to ensure that the isolated microaggregates were water-stable, backwashed in a glass jar, oven-dried (50 °C) weighed and stored. The material on the 250 $\mu$m sieve was also backwashed in a glass jar, oven-dried (50 °C) weighed and stored for further analysis.

## 2.4 Size fractionation of SOM to MAOM and POM and sand correction

A size fractionation approach was used to distinguish SOM to its components, i.e., MAOM and POM as described in Cambardella and Elliott (1992), and to determine the sand content of all aggregate fractions >53 $\mu$m using the method of Elliott et al. (1991). Specifically, the average upper size limit specification from MAOM at 53 $\mu$m was used which is also considered as the lower limit for the sand fraction. An aliquot of 5 g from each aggregation fraction was dispersed in 20 ml sodium hexametaphosphate (5 g L$^{-1}$), (0.5 % ) for 18 hours at 180 rpm, and sieved through a 2 mm sieve to remove possible small stones and a 53 $\mu$m sieve to determine sand content. The fraction remaining in the latter sieve was backwashed in a glass jar, oven-dried (105 °C) until constant mass and weighed. The procedure was replicated twice for each sample and the average value was used for the following analysis. By dispersing and sieving the fractions through the 53 $\mu$m sieve, the fraction remaining in the sieve was considered sand and the POM component of the SOM and the fraction that passed through the sieve was taken as the MAOM component and s&c fraction.

The relevant aggregate weight percentages were corrected for their sand content to allow comparisons between soils with different sand contents. The sand corrected fraction proportions were calculated as proposed in Six et al. (2002).





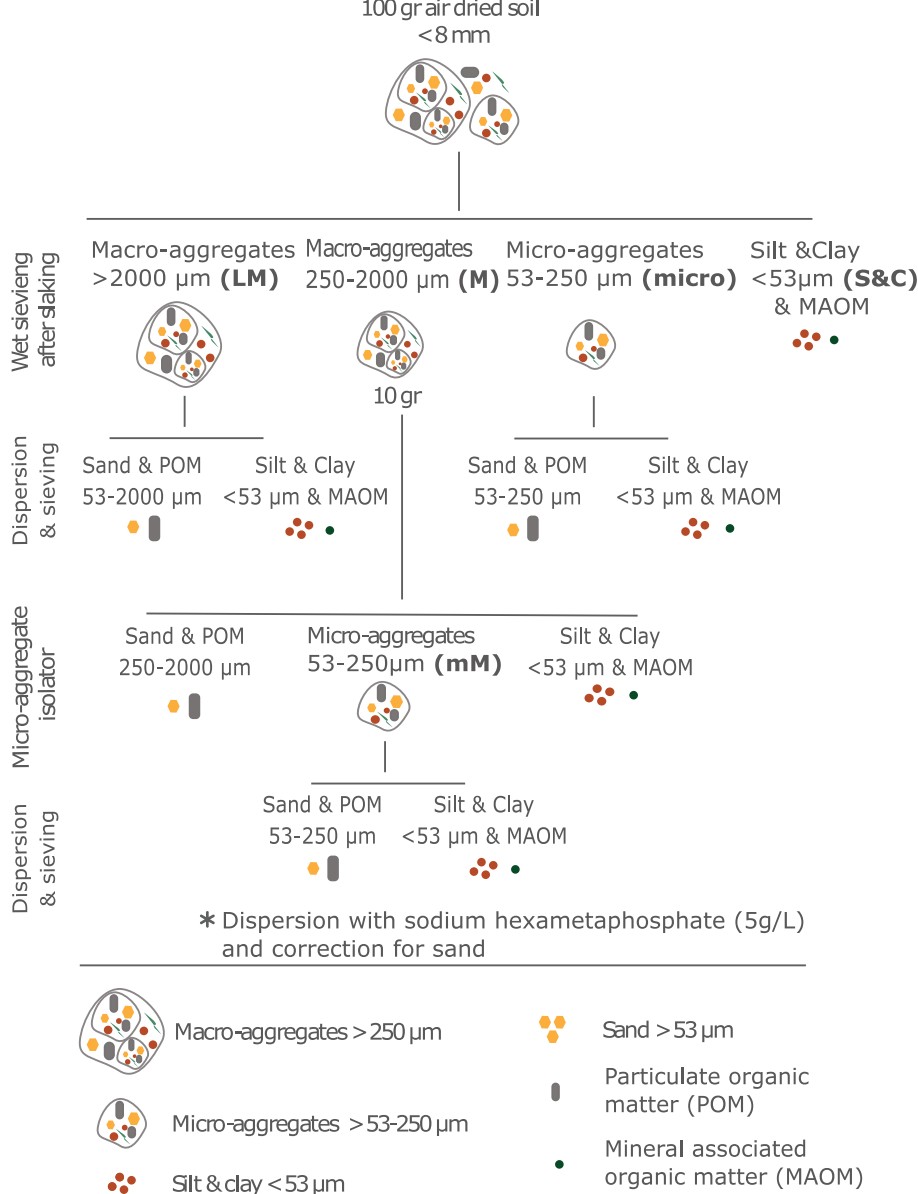

**Figure 1.** Aggregate and SOM separation scheme used in this study. For each fraction the different soil particles and relevant organic matter components as considered in this research are presented. This separation scheme designed following a combination of methods and concepts described in Cambardella and Elliott (1992); Elliott et al. (1991); Six et al. (2000a)

The soil stability was assessed using the Mean Weight Diameter (MWD) (Eq. 1) which is the most widely used index, defined as the sum of the weighted mean diameter of the aggregate size fractions where the weighting factor of each fraction is the proportion of the total sample weight for each fraction (Nimmo, 2013).



$$MWD\,(\mu m) = \frac{8000\,\mu m + 2000\,\mu m}{2}\,LM\% + \frac{2000\,\mu m + 250\,\mu m}{2}\,M\% + \frac{250\,\mu m + 53\,\mu m}{2}\,micro\% + \frac{53\,\mu m}{2}\,s\&c\% \quad (1)$$

## 2.5 Soil Organic Carbon determination

Soil organic carbon content (SOC) which is the largest and easiest component of SOM to quantify, was used in this study. The SOC content was determined by dry combustion and mass spectrometry elemental analysis (Carlo-Erba EA 1110, Thermo Scientific). A representative subsample of each fraction was taken with a soil sample splitter, ground and weighed into an Ag capsule. To determine only the carbon present in organic form, carbonates were removed with the addition of HCl (35 %). After drying at 40 °C for 24 hours, the soil samples were loaded into the autosampler for combustion with oxygen with the presence of chromium trioxide (catalyst). The mass percentage was determined after quantification by infrared absorption spectroscopy of the organic carbon (OC) that reacted to carbon dioxide ($CO_2$).

## 2.6 Statistical analysis

One-way Analysis of Variance (ANOVA) (Webster, 2007) was carried out using R-Studio with R version 3.6.1 (R Core Team, 2019; RStudio Team, 2016) to test for differences between treatments. Estimated marginal means by factors were computed by the least square method using the package "emmeans" (Lenth, 2020). All graphs were produced with the package "ggplot2" (Wickham, 2016). The statistical significance was checked at p <0.05. The assumptions of normality and homoscedasticity of the residuals were assessed by visual inspection of the Q-Q plots and plots of the normalized residuals against the fitted values.

# 3 Results

## 3.1 Soil stability

The soil structure stability of the sand corrected aggregates represented by the MWD (Figure 2 ) was a sensitive index reflecting management changes. In CZ, the zero-tillage treatment had almost double MWD in comparison to the minimum or conventional tillage. The same applied to the HU_2 experiment between the shallow and conventional tillage. The UK experiment presented the highest MWD values of all the experiments monitored up to 3000 $\mu$m for the direct drilling (zero tillage) treatment, which was significantly higher than the conventional tillage.

The addition of animal manure or compost led to higher MWD but differences among the treatments were only statistically significant in the BE experiment. Adding 45 Mg ha[-1] of VFG compost annually during a long period, led to a MWD of about 1200 $\mu$m which was significantly higher compared to the addition of 45 Mg ha[-1] tri-annually (one third of the previous treatment) and from no addition of compost. In the IT_1c, IT_1p and HU_1 experiments, there were no significant differences among the treatments but the visual trends were consistent. Adding manure increased the MWD. In the HU_1 experiment the incorporation of the crop residues led to significantly higher MWD compared to the NPK-only treatment. Similarly, in





the IT_2c and IT_2l experiments, the incorporation of residues resulted in higher values of MWD, but the differences were statistically significant only in the IT_2l.

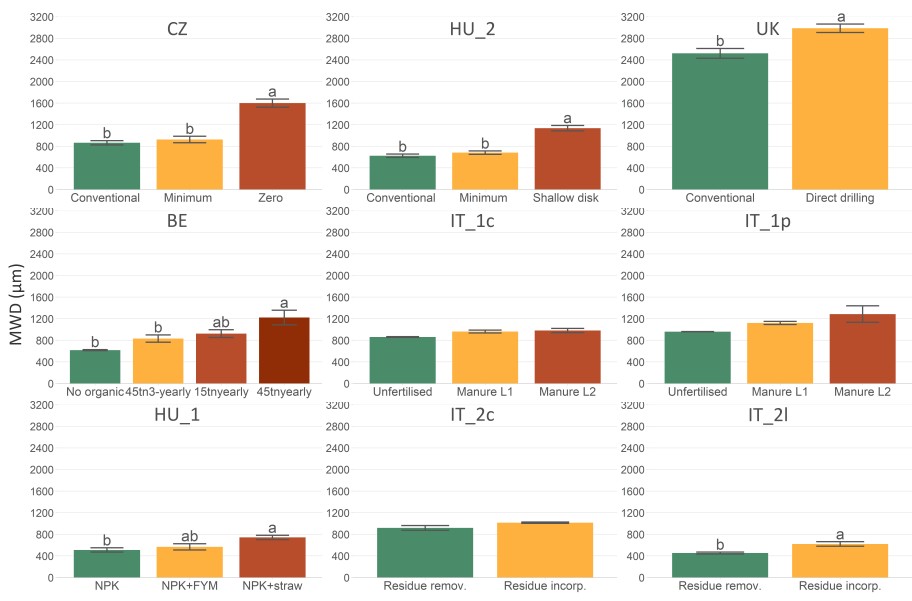

**Figure 2.** Mean weight diameter (MWD) of sand free aggregates in the topsoil (0–15 cm) for each study site (see also Table 1 for the codes and a description of the treatments). Error bars represent the standard error. Within an experiment, bars with a different letter differ significantly according to Tukey's test (p<0.05).

### 3.2 Soil Organic Carbon

The total SOC (Fig. A1) expressed in its two main fractions (i.e., POM and MAOM) (Figure 3), ranged between 10 g C kg$^{-1}$ soil and 70 g C kg$^{-1}$ soil (1 % –7 %) among all experiments. In all cases when significant increases in the total POM were

identified as a result of management, the total SOC was also increased. The treatments with the minimum soil disturbance in CZ and HU_2 and the treatments where manure or compost were applied in BE, IT_1c, presented significantly higher values of total POM and MAOM in the topsoil, than the treatments where conventional inversion tillage took place or when they were unamended. The retention of the crop residues did not lead to an increase of the topsoil MAOM when compared to their removal while the incorporation of straw significantly increased the POM in HU_1. Even if no significant differences were

observed, the IT_1p experiment where an initially peaty soil was treated with manure presented the highest values both of POM and MAOM with the latter being as high as 45 g C kg$^{-1}$ soil.

### 3.3 Aggregate weight and organic carbon distribution

Reduced tillage intensities caused statistically significant differences in the mass distribution of water-stable large macroaggregates and microaggregates, with those causing less soil disturbance resulting in more water-stable large macroaggregates but





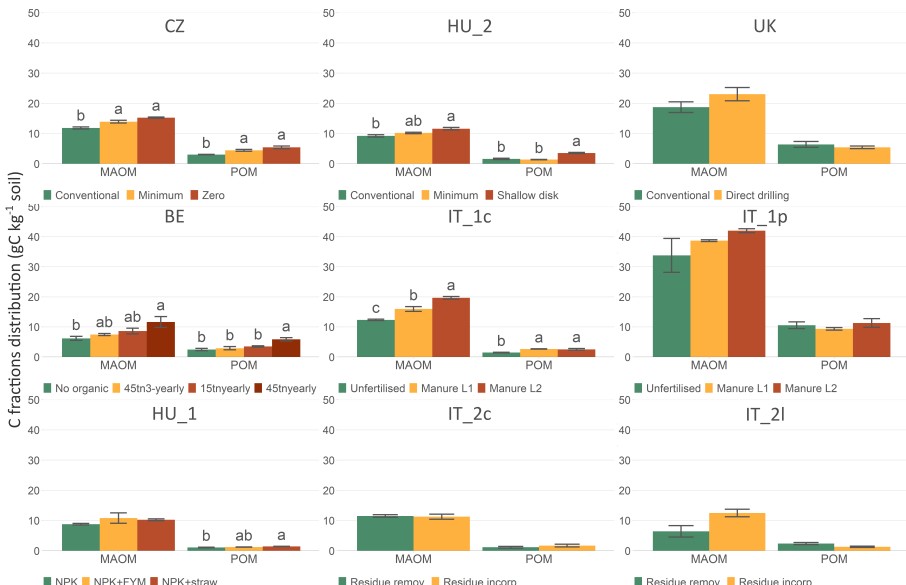

**Figure 3.** Total topsoil (0–15 cm) SOC separated in its components for each study site (see Table 1 for the codes and a description of the treatments). MAOM stands for mineral associated organic carbon and POM for particulate organic carbon. Error bars represent the standard error. Within an experiment and fraction, bars with a different letter differ significantly according to Tukey's test (p<0.05).

fewer microaggregates (Figure 4). The macroaggregates between 250 $\mu$m and 2 mm did not present significant differences in the CZ and HU_2 experiments, but they differed significantly in the UK experiment in which the conventional tillage resulted in more water-stable macroaggregates than the direct drilling treatment. Less intensive practices resulted in a small but statistically significant decrease in the s&c fraction in the UK and CZ experiments. Different rates of manure or compost applications caused significant differences (in case of BE and IT_1p) or an increasing trend when the organic input level was higher in the

large macroaggregates. In the BE experiment, an abundance of water-stable microaggregates was also observed in the treatment without addition of compost compared to those with addition. In the HU_1 experiment, the application of FYM resulted in more macroaggregates and fewer microaggregates than when only mineral fertilizer was applied. In addition, when the straw was incorporated in the soil, we found significantly more water-stable macroaggregates and less microaggregates than when FYM and mineral fertilizer or only mineral fertilizer was applied. The residue retention in IT_2c and IT_2l did not result in

significant differences (except the microaggregates fraction in IT_2l) in the aggregate size fractions but they did present the same trends as the HU_1 experiment (i.e., water-stable large macroaggregates and macroaggregates and fewer water-stable microaggregates when the crop residues were incorporated in the soil than when they were removed). In all the other aggregate fractions not specified, there were no statistically significant differences among the treatments.

The SOC content in each aggregate fraction after the sand correction is illustrated in Figure 5. In most of the tillage exper-

iments the aggregates OC content in the upper 0–15 cm of soil was significantly higher when soil was not inverted compared to when it was. In the CZ experiment, zero tillage resulted in higher OC contents than conventional tillage in every aggregate



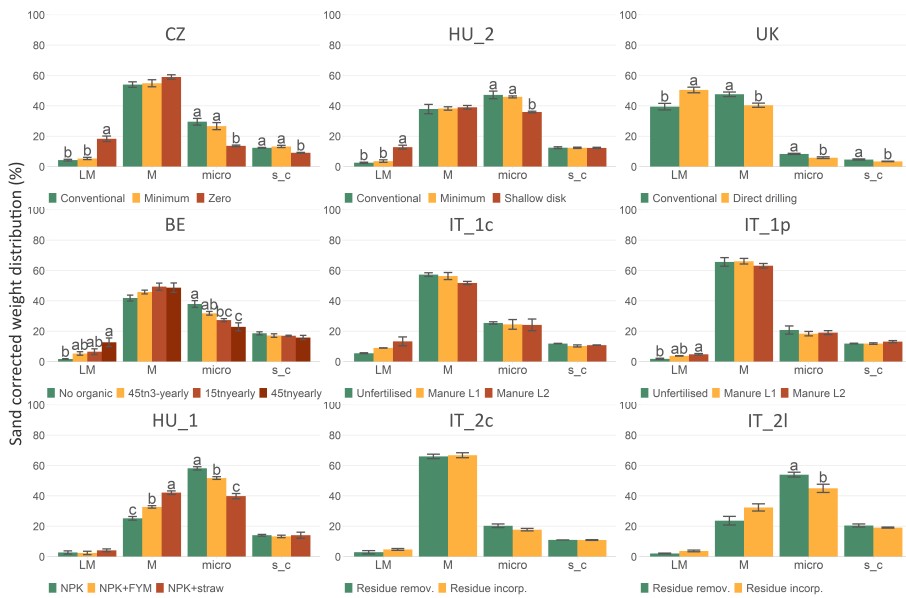

**Figure 4.** Weight distribution of sand free aggregate fractions for each study site (Codes and description of the practices are presented in Table 1). The error bars represent the standard error. Within an experiment and fraction, bars with a different letter differ significantly according to Tukey's test ($p < 0.05$).

fraction except the large macroaggregates, although this fraction exhibited the highest OC content up to 35 g C kg⁻¹ soil. Interestingly, the minimum tillage resulted in the same OC content with the zero tillage in the macroaggregates fraction. In the microaggregates fraction, minimum tillage presented significantly lower values than the zero tillage and significantly higher

OC content than the conventional tillage, and in the s&c fraction, significantly lower carbon content than the zero tillage was observed. In general, the HU_2 experiment followed the same trend as the CZ. Shallow disking resulted in statistically higher OC than conventional tillage in all fractions apart from the large macroaggregates, but OC contents under minimum tillage did not differ from conventional tillage. In the UK experiment, significant differences between the two treatments were only present in the microaggregates and s&c fractions, with the direct drilling presenting higher OC content (more than 40 g C kg⁻¹

soil in the microaggregates fraction).

In the compost or manure experiments, i.e. BE, IT_1c and IT_1p, the OC content in the aggregate fractions (Figure 5) responded as the total OC (Figure 3), meaning that higher application rates led to higher OC contents in the different fractions. In the BE experiment, the annual application of 45 Mg ha⁻¹ compost led to significantly higher OC in the macroaggregates and microaggregates fractions compared to every other treatment, whereas no significant differences existed among the rest

of the treatments. In the IT_1c experiment, addition of manure led to significantly higher OC among all the treatments in the macroaggregates fraction, and between the Manure_L2 and unfertilized treatment in the s&c fraction. In the microaggregates and large macroaggregates fractions, there was large variability between the experimental plots and no significant differences were found. Similarly, in the IT_1p experiment, where even though the OC content within the different fractions went up to





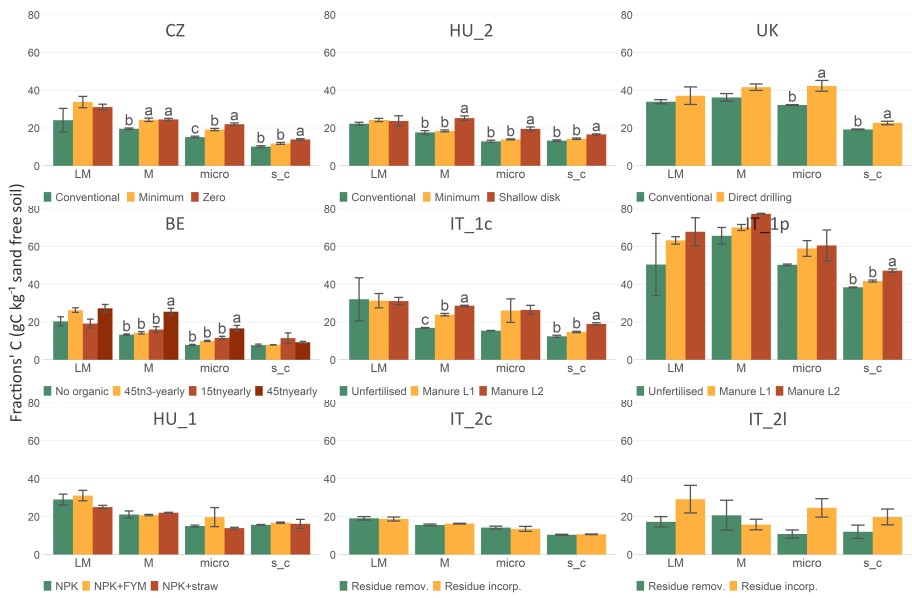

**Figure 5.** SOC content in each aggregate fraction expressed in a sand-free soil for each study (Codes and description of the practices are presented in Table 1). The error bars represent the standard error. Within an experiment and fraction, bars with a different letter differ significantly according to Tukey's test (p<0.05).

about 80 g C kg⁻¹ soil, the in-between experimental plots variability did not allow for safe comparisons among the treatments. In the experiments where the treatments dealt with the residues management, i.e., HU_1, IT_2c and IT_2l, we did not observe significant differences among the treatments.

### 3.4 Mass and OC dynamics in the occluded microaggregates

Based on our observations, we found that tillage intensity and OM additions i) influenced the quality of macroaggregates in terms of OC content, but not their mass, and ii) led to a negative relationship between OC content and aggregate mass in the microaggregates fraction. We therefore isolated the microaggregates within macroaggregates (mM) to understand better the underlying processes. The analysis did not show significant differences among the treatments in all categories on the amount of mM (data shown in Fig. A2), but it did show significant differences in their OC content (mM-C) (Figure 6). Over all study sites, their OC content ranged from 10 to 47 g C kg⁻¹ mM. In the CZ experiment significantly more mM-C was observed in the zero tillage in comparison with the conventional, while in the HU_1 there was significantly more occluded OC in the shallow disking +secondary tillage compared to the other two treatments (i.e., shallow disking alone and conventional deep ploughing). In the UK experiment, OC content in the mM fraction did not differ among the treatments. In the BE experiment, only the annual addition of 45 Mg ha⁻¹ of compost led to increased OC content in the mM fraction compared to the other application rates. Both in the IT_1c and IT_1p experiments, the occluded OC was higher when manure was added than when




it was not, but the difference was statistically significant only for IT_1c. The residue retention led to more mM-C only in the
HU_1 experiment.

In all experiments the MAOM dominates in the mM fraction when compared to the POM (Figure 7). When comparing the
differences in each OC component among the treatments the same trends with the total OC were followed/observed. In the
BE experiment when 45 Mg h$^{-1}$ of compost were added the MAOM was double compared to the treatments with no addition
of compost and statistically significantly higher than the other two treatments with lower amounts of compost addition. In the
IT_1c experiment when animal manure was added (both levels) the MAOM was significantly higher than in the unfertilized
treatment. The same results were observed in the HU_1 experiment. Comparing the treatment that included only application
of mineral fertilizers with those that got either manure or residues, we observed that the MAOM was significantly higher in the
case of farmyard manure addition and retention of the residues.

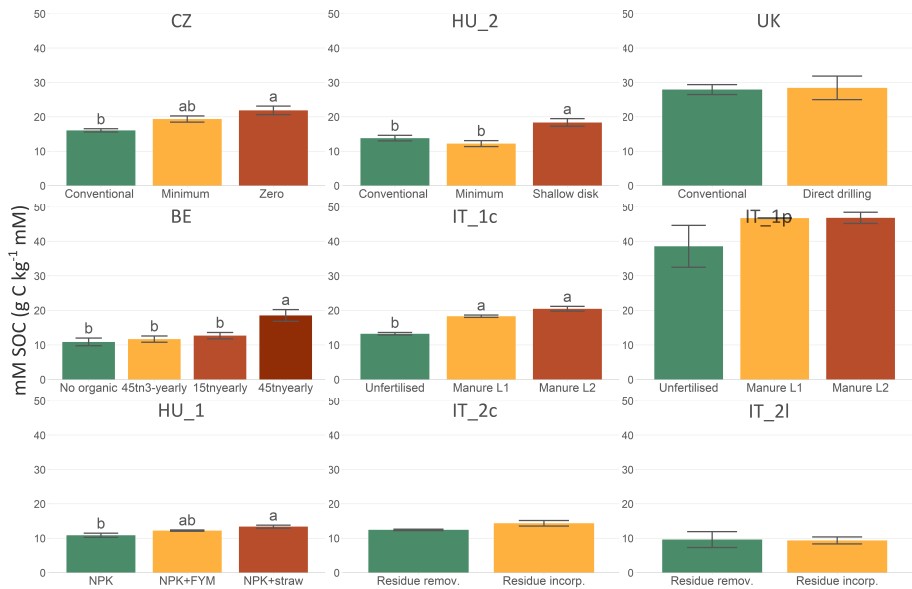

**Figure 6.** SOC content in the microaggregates held within the macroaggregate fraction (mM-C) (Codes and description of the practices are
presented in Table 1). The OC content is expressed in g C per kg of occluded microaggregates. The error bars represent the standard error.
Within an experiment, bars with a different letter differ significantly according to Tukey's test (p<0.05).

### 3.5   Crop yield

The different treatment categories did not present specific patterns or consistent differences when it came to crop yield (Fig.
A3). In the tillage experiments, only the treatments in the CZ experiment caused statistically significant differences. Specifi-
cally, both in 2019 and 2020 the minimum tillage yielded more than the conventionally tilled field. Zero tillage did not cause a
yield loss when compared with conventional tillage. Addition of manure caused a statistically significant increase in the IT_1c
and the IT_1p (only in 2019) when compared to the unfertilized plots. Addition of compost in BE did not cause significant

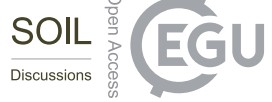

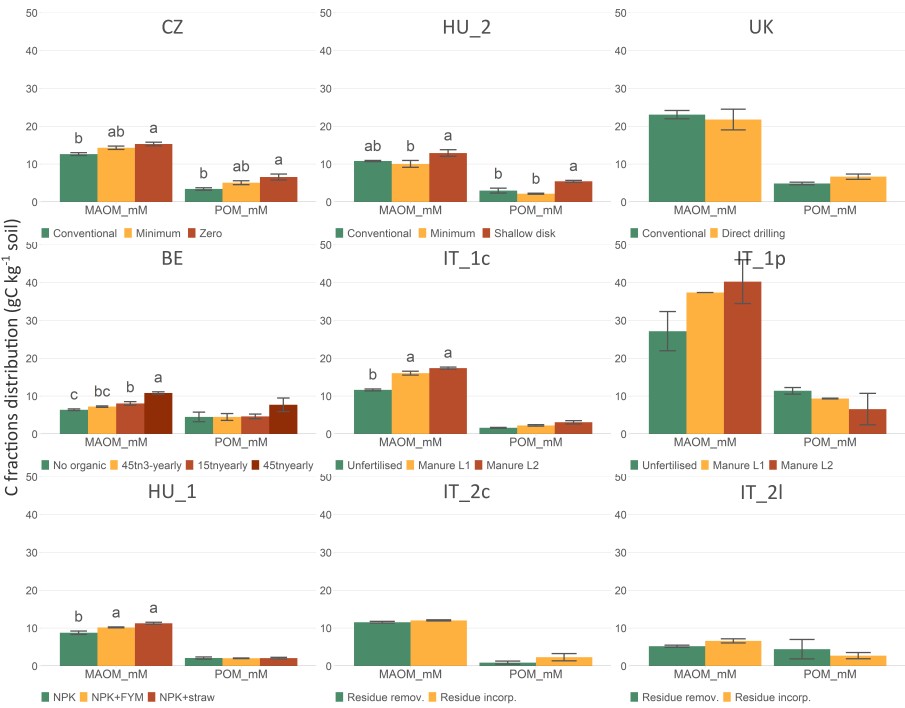

**Figure 7.** OC content in each SOM component in the mM size fraction in each study site (Codes and description of the practices are presented in Table 1). The OC content is expressed in g C per kg of occluded microaggregates. The error bars represent the standard error. Within an experiment and fraction, bars with a different letter differ significantly according to Tukey's test ($p<0.05$).

differences in yields. Finally, the incorporation of residues created only a small increase in individual cases such during 2019 in HU_1 and 2020 in the IT_2l experiments.

## 4    Discussion

In this study, we explore the important challenge (Dignac et al., 2017) of how to increase topsoil SOM and improve the topsoil structure through improved soil management while ensuring agricultural production and farmers' income by comparing our results with available literature. Restoring degraded agricultural soils via natural attenuation (i.e. set-aside) demands long periods of fallow, especially in regions with low primary production (Apostolakis et al., 2017), which is not compatible with ensuring farmers' income or societal objectives for food production. This is especially important for regions with stagnating yields or with yields that are considerably less than their potential (Wiesmeier et al., 2015; Schils et al., 2018) due to land mismanagement and poor soil quality, and for regions with low primary production, such as those of the Mediterranean basin, where low C inputs into the soil limit aggregation and OC sequestration (Apostolakis et al., 2017). For this reason, additional measures are needed.





Common practices include organic and/or mineral fertilization, addition of soil amendments such as compost or manure, management of crop residues and various tillage intensities (from conventional to zero). These practices control both the spatial and temporal distribution of OM inputs into the soil and regulate the sensitivity of OM to mineralization, affecting
with these mechanisms the OM stocks (Jastrow et al., 2007). Such practices can i) reduce OM mineralization by minimizing disturbances, ii) promote aggregation and iii) increase the SOC content and aggregate stability, but on the other hand they might also enhance the decomposition of old OC due to the addition of fresh OM inputs (e.g. residues), i.e., the priming effect (Fontaine et al., 2007; Liu et al., 2020).

Soil tillage by inversion ploughing is a common agricultural practice which, despite being associated with multiple benefits,
is often held responsible for the deterioration of soil structure, imposing severe degradation threats (FAO, 2003; Karlen et al., 1994b, 2013) including SOM losses (Six et al., 2000a). This happens directly because of the destruction of soil aggregates and the deterioration of the SOM physical protection and/or indirectly by exposing the modified structure to wetting- drying (or freezing - thawing) conditions as well as rainfall (Al-Kaisi et al., 2014; Balesdent et al., 2000; Paustian et al., 1997). Most studies did not investigate the effects of tillage on soil structure and SOM distribution in various aggregate fractions
from multiple regions and under a common methodological framework, which leads to strong controversies in the literature (Blanco-Canqui and Lal, 2008; Haddaway et al., 2017; Karlen et al., 2013; Sheehy et al., 2015). To test our hypotheses and to evaluate the potential to reverse the negative effects of inversion tillage in the topsoil, we exploited a network of ongoing long-term experiments. These experiments are testing different, alternative to inversion tillage, practices for their effects on topsoil structure and OC as well as practices that deal with additions of different organic materials under conventional tillage.
Generally, it was observed that inversion tillage in CZ, HU_2 and UK and less organic inputs in BE, IT_1, HU_1 and IT_2, deteriorated the topsoil aggregate stability and decreased the topsoil SOC. Moreover, inversion tillage and less organic inputs led to fewer large macroaggregates with no profound differences in the OC content within these, and mainly no differences in the macroaggregates quantity but a significantly lower OC content in these macroaggregates. Finally, these practices resulted in more microaggregates, but carbon depleted, and a small increase in the s&c fraction weight but with reduced OC content.
The results obtained after soil separation into aggregates and the determination of their OC content confirm our first hypothesis. Specifically, the inversion tillage reduced the mass of water-stable large macroaggregates (Figure 4) and the SOC stabilized in all fractions (Figure 5) including the mM (Figure 6). However, we did not observe a decrease in the mass of macroaggregates under inversion tillage as also observed in other studies (Modak et al., 2020; Piazza et al., 2020), especially in CZ and HU_2. The tillage systems that reduced soil disturbance to a minimum or zero level and avoided excessive soil inversion, (e.g., CZ,
UK and HU_2), improved the topsoil stability, increased the mass of large macroaggregates, did not affect the mass of the macroaggregates and reduced the mass of all other fractions and finally increased the OC in all fractions in the 0–15 cm soil layer. The mass decrease of the large macroaggregates and the reduction of the SOC in the aggregates fractions in conventionally tilled soils has been presented in several studies (Devine et al., 2014; Jat et al., 2019; Mikha and Rice, 2004; Sheehy et al., 2015) but most of them also found a reduction of the macroaggregates mass (Jat et al., 2019; Mikha and Rice, 2004; Mondal
and Chakraborty, 2022; Plaza-Bonilla et al., 2010; Song et al., 2019; Zheng et al., 2018). As this was not found in our results



(all experiments except for the UK where reduction occurred in the mass of large macroaggregates), the composition of the macroaggregates was evaluated further in our work.

Our second hypothesis dealt with the addition of exogenous material which already underwent decomposition before the application on the soil, in fields in BE, IT_1c, IT_1p and HU_1 where conventional inversion tillage was the main soil culti-
vation practice. Under most conditions, adding exogenous organic material such as compost or manure increased substantially the OC content within all aggregate fractions as well as the total OC in the topsoil , even if combined with inversion tillage (De Clercq et al., 2016; Lin et al., 2019; Mikha and Rice, 2004; Zhang et al., 2021). Interestingly though, we did not observe an increased mass of water-stable macroaggregates as initially hypothesized and demonstrated in previous studies (Mikha and Rice, 2004; Wen et al., 2021; Wortmann and Shapiro, 2008). Results similar to ours, were obtained by Lin et al. (2019) and Zhang et al. (2021) who even observed a reduction of the macroaggregates and an increase of the large macroaggregates after manure fertilization. Nevertheless, addition of exogenous organic material increased the mass of the large macroaggregates as also observed by Mikha and Rice (2004) which, despite being a small fraction, is important for maintaining a good soil structure.

While exogenous sources had a pronounced impact on soil structure and SOM regardless of the tillage intensity as also observed by Gross and Glaser (2021), the effects of crop residue left on the soil or incorporated into the soil varied among the different study sites. Based on our findings in all experiments that dealt with residue management, our third hypothesis cannot be supported. Even if the stability of the soil improved, with several experiments to present statistically significant increase in the MWD as also observed by Modak et al. (2020) and mass of macroaggregates, the corresponding SOC contents as well as total SOC were not affected positively by the incorporation of fresh organic materials as had been reported by Powlson et al. (2011). Similar results were obtained by Lin et al. (2019) while other studies demonstrated a parallel increase of aggregates mass (large macroaggregates, macroaggregates and microaggregates) and respective OC content (Fuentes et al., 2012; Karlen et al., 1994a; Zhao et al., 2018). This may be because the exogenous organic materials (compost in BE or manure in IT_1) were at least partially decomposed and so in a more stable form. In contrast, the crop residues were not previously decomposed, and they decomposed fast when applied to the soil (Struijk et al., 2020). Berti et al. (2016) previously showed how the greater humification coefficient might be responsible for greater SOC stock under manure application compared to crop residue incorporation, with the coefficient of manure (0.1588) to be almost three times greater than the one for the crop residues.

In addition, there are concerns and evidence (Dignac et al., 2017; Fontaine et al., 2007; Wang et al., 2015) that the addition and incorporation of fresh organic material throughout the soil profile with conventional deep ploughing may trigger the microbial demand for carbon. This may lead to increased mineralization and consumption of the old existing OC in the soil (i.e., priming effect). Nevertheless, the priming effect, input quality and decomposition rate seem to play a role only in the short term as in the long-term there is evidence from Cardinael et al. (2015) and Thomsen et al. (2013) that do not show differences in the SOC retention, regardless of the initial organic material quality. The combination of residue retention with zero (Pu et al., 2019), minimum tillage, or shallow ploughing could be more effective for SOC storage increase even in the short term



compared to conventional tillage practices (Dal Ferro et al., 2020; Fuentes et al., 2012; Li et al., 2020; Luo et al., 2010; Xu et al., 2019).

Reduced or no-tillage keeps fresh organic material concentrated on the soil surface or in the topsoil (Andruschkewitsch et al., 2014; Virto et al., 2012) and keeps the roots intact increasing the POM concentration. Commonly, the roots contribute more to SOC storage than the above-ground inputs, due to an approximately double efficiency in conversion into stable SOC

compared to above-ground biomass (e.g., crop residue) (Berti et al., 2016; Kätterer et al., 2011; Rumpel and Kögel-Knabner, 2011). Reduced topsoil disturbances promote aggregation and subsequently provide adequate time for the added fresh POM to become protected within large aggregates and further to become stable in the occluded microaggregates (Six et al., 2000a). These processes lead to reduced mineralization not only in the zero tillage systems but also under reduced tillage in several cases (Chen et al., 2019; Ghimire et al., 2017). The concentration of the residues in the topsoil and the lack of mixing of these

residues throughout the soil profile have been linked to constraints regarding the maintenance of OC level in the deeper soil horizons. Some authors claim that the SOC balance over the whole soil profile remains unchanged (Blanco-Canqui and Lal, 2008; Chen et al., 2019; Haddaway et al., 2017). Nevertheless, there is recent evidence that the adoption of zero or reduced tillage systems also maintains better the existing SOC levels in the deeper horizons (Cooper et al., 2021; Varvel and Wilhelm, 2011), so, increasing the levels of SOC in the topsoil and keeping the subsoil SOC content unchanged, lead to a net SOC

increase in the system (Cooper et al., 2021).

Also, we can argue that even if the effects of reduced and zero tillage are limited when it comes to SOC increase in the soil profile, the improvement of the structural stability expressed with the MWD in this research is maybe more important, as several other soil functions are improved (Rabot et al., 2018). The same applies for the limited effects on topsoil OC of crop residues retention (Panagea et al., 2021). The concentration of the residues in the topsoil and on soil surface mostly influence

aggregation (Zhang et al., 2014), and it is beneficial to reduce soil erosion, and prevent soil crusting, increase infiltration and water retention, reduce evaporation and enhance the earthworm density leading to overall better topsoil quality (Busari et al., 2015; Gao et al., 2017; Panagea et al., 2021; Rabot et al., 2018).

An important aspect is to unravel the soil processes responsible for long-term OC storage and, more importantly in the agricultural sector, the SOC components that are sensitive to management changes. This allows identification of the changes of

management effects, which could be either positive or negative towards a desired soil composition and structure. The physically protected OC within the microaggregates fraction has been identified as the second most protected after the OC chemically bonded to the soil mineral particles. A linear negative relationship between the microaggregates fraction weight percentage and their carbon content was detected in this study (F-stat: 15.61, p-value: 0.000171), as also found in agricultural or native soils (Six et al., 2000b) under different levels of disturbances and organic material addition. This points out that the increasing

cultivation intensity and the lack of organic materials lead to a loss of carbon rich macroaggregates and to an increase of microaggregates which are though, C-depleted.

In most of the investigated experiments, the mass of both the macroaggregates (except for UK and HU_1) and occluded microaggregates were not affected by the different management practices or their intensities whereas, total SOC (g C kg$^{-1}$ soil), macroaggregate C and mM-C (g C kg$^{-1}$ mM) were in most cases positively affected when the tillage was reduced,



or when more organic materials were added. Therefore, we can partially reject our fourth hypothesis. Increased SOC and lower soil disturbances led indeed to more mM-C but not necessarily to more water-stable mM. The parallel change of the occluded microaggregates with the corresponding OC content (i.e., more mM-C when the mM are increasing) is proposed by Six et al. (2000a) and supported by several other studies (King et al., 2019; Modak et al., 2020; Piazza et al., 2020; Zhao et al., 2018). However, our findings do not support this hypothesis. Similarly, Andruschkewitsch et al. (2014) and Denef

et al. (2007) also concluded that there is no concomitant change in specific cases. Thus, the relationship between mass of occluded microaggregates and corresponding OC as well as the applicability of this concept in different soils and under various management practices should be further evaluated.

In the experiments with significant differences in the mM-C (g C kg$^{-1}$ mM) (Figure 6), the differences were reflected by the different occluded carbon components, i.e., mM-MAOM (g MAOC kg$^{-1}$ mM) and mM-POM (g POC kg$^{-1}$ mM) (Figure 7), as

well as, the total SOC content (g C kg$^{-1}$ soil) (Figure 3). In the treatments in which the soil was less disturbed compared to those that were under conventional tillage, both the MAOM and POM were higher, as also found by Modak et al. (2020). Taking into account that the quality of the macroaggregates and the occluded microaggregates depends on the POM (Six et al., 1998), which is trapped initially within the macroaggregates, together with the above-mentioned results on their mass we suggest that the total soil OC content is independent of the macroaggregates quantity but rather on their quality in terms of POM which

is trapped within them. In the experiments which included addition of organic materials like manure or compost, an increase was found only in the MAOM component. In contrast, in the experiments that incorporated the fresh residues in the soil no changes in the MAOM nor in the POM were identified. POM is mainly plant derived and has been proven in other studies to increase when the residues are retained or incorporated in the soil (Modak et al., 2020) or if manure is applied (Wen et al., 2021). In our research these practises were combined with inversion conventional tillage, and thus we suggest that the time that

the macroaggregates remain intact is crucial for the labile POM to stabilize within the occluded microaggregates.

An important difference found between exogenous (matured) material and fresh crop residue in our results is that the first can possibly counteract the inversion tillage effects on topsoil OC while the latter cannot. In the case of matured exogenous organic material, as in BE there is an increasing trend of SOC with increasing compost doses under conventional tillage, which is visible throughout all treatments but is only statistically significant at the highest dose. This does not mean that there is no

effect at lower doses. However, because of the great field variability, statistical differences are often difficult to demonstrate. That is the reason that the extremely high dose for Belgium of 45 Mg ha$^{-1}$ compost to the plots every year has been included even if from a practical point of view, it is not feasible to be implemented in large scale agricultural fields. In the case of IT_1 both levels of manure application in the clay soil increased the total soil OC when compared with the unfertilized field (no Nitrogen of any nature). Again, the option of not fertilizing the fields is neither preferred nor common in the current

agricultural systems. It should be stressed that, even if fields under mineral fertilization (NPK fertilization but no manure) had been compared with fields that received manure, differences in the OC content probably will not be noticeable, as in Italy the mineral fertilization alone seems not able to maintain the SOC levels (Lugato et al., 2010).

Based on the above-mentioned results, we agree partly with the suggestion of Chenu et al. (2019) that the best option to increase SOC is to increase the inputs compared to decreasing the outputs through mineralization. However, we need to stress

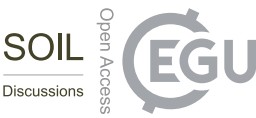

that a combination of increased organic inputs coupled with less intensive tillage methods that will stop or reduce the further
release of SOC seems more optimal and realistic. Adding the parameter of reduced tillage intensity, the time required for the
microaggregates to be formed properly within the macroaggregates and the OC to get stabilized within these is considered. In
addition, most probably by reducing the depth and intensity of tillage, the priming effect will be weakened in systems with
addition of fresh OM, leading to higher SOC stocks. This proposal should be validated with further studies including factorial
treatments of several tillage intensities and different organic materials, ideally over broad environmental conditions.

Finally, there is no simple answer/solution for increasing topsoil OC and aggregate stability and improving crop yield by
management practices. None of the practices evaluated in this research caused a yield reduction opposing the results of Song
et al. (2019), and this offers opportunities for a possible adoption from farmers, but our results also showed that there is
a small potential for adopting practices that improve the topsoil SOC and stability, to increase the crop yields. Individual
similar field experiments have shown a positive relationship between SOC and crop yield (D'Hose et al., 2014), however
similar to our results Vonk et al. (2020) found a poor association between yield and SOC for common crops in Europe and
highlighted the need for a diversified strategy when it comes to motivating farmers to increase their field SOM. The selection
and adoption of amelioration practices for a region depends on a variety of sociocultural, economic, biophysical and personal
factors (Bijttebier et al., 2018; Sattler and Nagel, 2010) as well as access to information (Rust et al., 2021). Adopting new
practices often require increased short-term costs for investments in new machinery, training for farmers and deviations from
traditions, which remain major determinants and possibly more influential than long-term objectives for sustainability (Sattler
and Nagel, 2010). To acquire knowledge transferable to sustainable management and readily useful to farmers, studies should
focus on the effectiveness of new/improved practices on soil fertility, which are preferred and accepted by local farmers.

## 5 Conclusions

We analysed seven long-term experiments in Europe to evaluate soil management practices that could inverse the negative
effects of inversion conventional tillage on topsoil structural stability and consequently the SOC content. Following testing
several hypotheses we conclude that:

- Inversion tillage reduces the mass of water-stable large macroaggregates, and the OC stabilized in the macroaggregates
  and microaggregates and this effect increases with increasing tillage intensity, but it does not affect the mass of macroag-
  gregates.

- Addition of matured exogenous organic materials increases in most cases the soil OC content in all aggregate fractions
  even under inversion tillage, but it does not increase the mass of water-stable macroaggregates.

- Incorporation of the previous crops residues into the soil does not increase the mass of the water-stable macroaggregate
  nor the occluded POM and does not influence in a consistent way the OC content in the occluded or in free microaggre-
  gates.



- Increased organic material inputs and reduced soil disturbances do not lead to more water-stable occluded microaggregates but they do lead to increased OC within these (mM-C).

We propose that conventional inversion soil tillage affects topsoil structure in a two-fold way: i) mainly by disrupting the aggregation cycle, thus limiting the time for the development of a stable structure and, subsequently, the time for SOM stabilization within the occluded microaggregates and ii) by increasing the decomposition of OM, and thus limiting the OM that is available to promote aggregation. Therefore, we suggest that there are two ways to protect and/or mitigate soil degradation caused by conventional- inversion tillage in the topsoil: i) to reduce the tillage intensity and provide more time to the soil structure to build and to stabilize SOM and ii) to increase the OM inputs, combined with non-inversion tillage practices, in order to increase the SOC stocks and diminish the decomposition of old OC. By these ways, the topsoil structural stability, as well as the quality of the topsoil in terms of OC content improves with a subsequent positive effect on the topsoil functions.



## Appendix A

**Table A1.** Description of the study sites and experiments. Table taken from Panagea et al. (2021).

| Code | Town, Country | Coordinates (Decimal Degrees) | Agro-Climate Zone (Ceglar et al., 2019) | Start of Experiment | Soil Type | Name of Experiment | Reference |
|---|---|---|---|---|---|---|---|
| **CZ** | Prague-Ruzyně, CZ | 50.0880 14.2980 | Continental | 1995 | Silt Loam | Tillage trial | Mühlbachová et al. (2015) |
| **HU_2** | Keszthely, HU | 46.7346 17.2302 | Pannonian | 1972 | Silt Loam | Soil tillage systems in wheat and maize bi culture | Hoffmann and Kismányoky (2001) |
| **UK** | Loddington, UK | 52.6089 0.83257 | Maritime North | 2011 | Clay loam | Soil Biology and Soil Health | - |
| **BE** | Bierbeek, BE | 50.8244 4.79605 | Maritime North | 1997 | Silt Loam | VFG Compost trial | Tits et al. (2014) |
| **HU_1** | Keszthely, HU | 46.7332 17.2295 | Pannonian | 1983 | Silt Loam | Organic & inorganic fertilization trial- IOSDV | Kismányoky and Tóth (2013) |
| **IT_1c** | Legnaro, IT | 45.3506 11.9497 | Maritime South | 1964 | Silty Clay Loam | Organic & mineral fertilization trial | Giardini (2004) |
| **IT_1p** | Legnaro, IT | 45.3506 11.9497 | Maritime South | 1964 | Peat 18% OC initially | Organic & mineral fertilization trial | Giardini (2004) |
| **IT_2c** | Legnaro, IT | 45.3507 11.9498 | Maritime South | 1970 | Silty Clay Loam | Nitrogen fertilization and crop residue trial | Giardini (2004) |
| **IT_2l** | Legnaro, IT | 45.3507 11.9498 | Maritime South | 1970 | Silt Loam | Nitrogen fertilization and crop residue trial | Giardini (2004) |



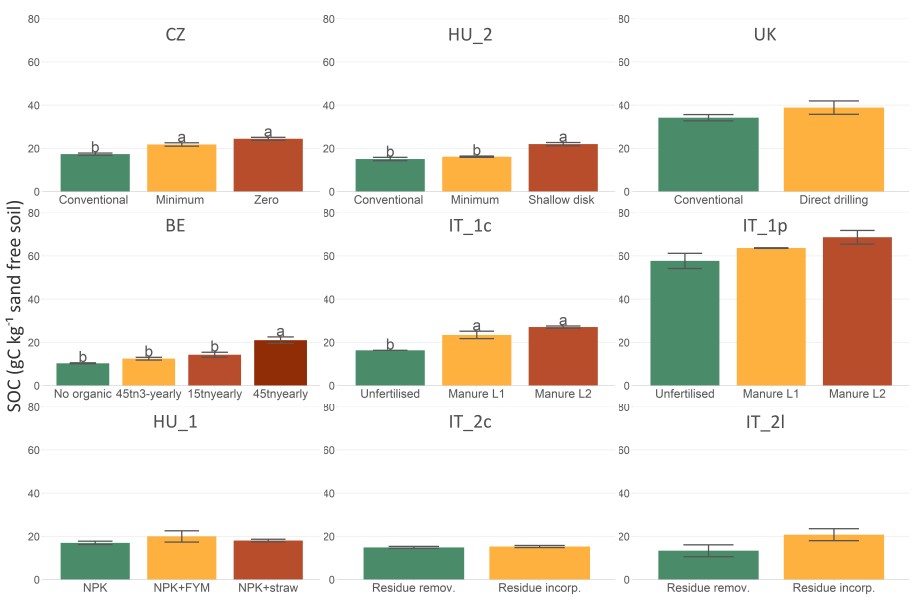

**Figure A1.** Topsoil (0–15 cm) organic carbon expressed in sand free soil for each study site. The OC content is expressed in g C per kg of sand free soil. The error bars represent the standard error. Within an experiment, bars with a different letter differ significantly according to Tukey's test (p<0.05).

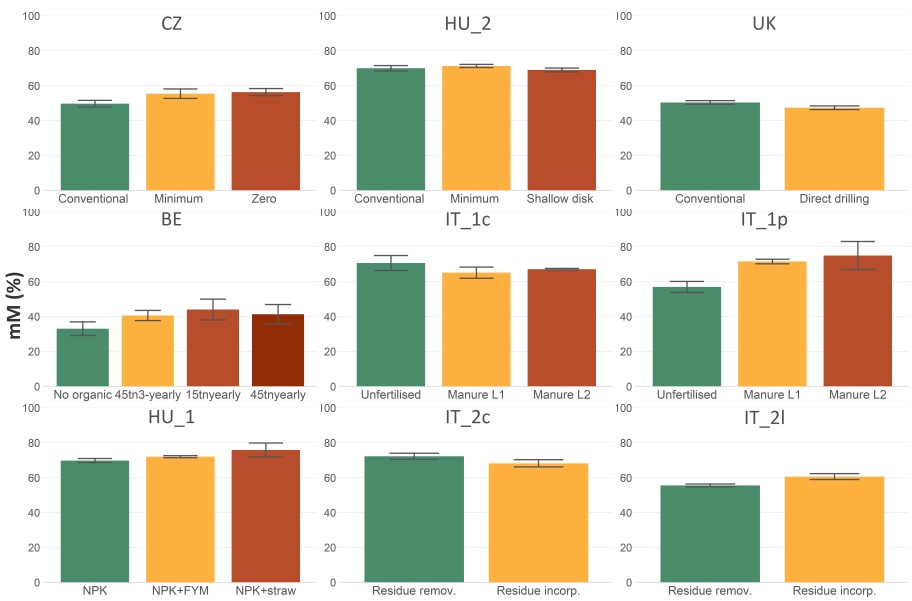

**Figure A2.** Mass percentage of the occluded within the macroaggregates, microaggregates. The mM mass is expressed as percentage of macroaggregates. The error bars represent the standard error. Within an experiment, bars with a different letter differ significantly according to Tukey's test (p<0.05).





**Figure A3.** Crop yield in each study site. The error bars represent the standard error. Within an experiment and year, bars with a different letter differ significantly according to Tukey's test (p<0.05).



*Data availability.*   All raw data will be available in the SoilCare community in Zenodo repository (https://zenodo.org/communities/soilcare,) after the acceptance of this paper.

*Author contributions.*   Conceptualization, I.S.P; methodology and soil sampling and analysis I.S.P.; data analysis, I.S.P.; visualization, I.S.P.;
writing—original draft preparation, I.S.P.; A.A. and G.W.; writing—review and editing: J.P.; J.D.; M.T., A.E., I.P., A.B., C.S., Z.T., H.K., J.B. and P.Č.; supervision, G.W.; J.D.; J.P.; project coordination, G.W.; funding acquisition, G.W.; All authors have read and agreed to the published version of the manuscript.

*Competing interests.*   The authors declare that they have no conflict of interest.

*Acknowledgements.*   We thank all the study sites owners for permitting the use of the experimental fields where this research was done and
providing all types of help during the sampling campaign. We thank the laboratory assistant of the Department of Earth and Environmental Sciences of the KU Leuven Lore Fondu and the postdoctoral researhcer Dr. Zita Kelemen for providing valuable guidance and help during the soil samples analysis.

*Financial support.*   This research was funded by the project "SoilCare: Soil Care for profitable and sustainable crop production in Europe" from the H2020 Programme under grant agreement no 677407.



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
