# Peer review of "Impact of agricultural management on soil aggregates and associated organic carbon fractions: Analysis of long-term experiments in Europe"

_SOIL, 2022_

## Author Comment (AC2)

We thank the reviewer for the detailed and constructive feedback. The remarks are useful to improve the readability of the manuscript, shorten all the sections and focus more on the important aspects of the research. Below we address all comments in detail. The comments by the reviewers are in red; our reply is in black

The study presented by Panagea and co-authors summarizes the outcomes of the analyses of topsoil structural stability as determined by soil aggregates, and related soil organic carbon under different soil managements that are proposed beneficial for soil (structural) quality. The authors made use of a European network of long-term field experiments, which allows them to cover a wide range of region-specific management practices, as well as pedo-climatic conditions. The approach is very good and the authors collected a valuable dataset. I would think that the authors should highlight even stronger the European gradient, which is both west-east and north south.

Thank you very much for appreciating the value of the dataset. A map will be included in the text to visualize the European gradient of the research as also suggested by Reviewer 2. Indeed, mentioning the coordinates in a table is not so informative for a reader. The statement in the Site description section will be changed: "The long-term experiments were set up independently from one another with different objectives and under different environmental conditions. Nevertheless, they offer the possibility to explore a wide range of representative management practices and pedo-climatological conditions across Europe"

[Figure]

The manuscript is generally nicely written and understandable. Some sentences are long and should be split into two. For example, ll. 344 – 346 should be split into two sentences, one stating the outcomes of the study and the second one highlighting that this has also been observed in other studies. Please avoid long sentences containing too much information.

The specific sentence was rephrased and shortened as well as several other sentences were shortened or split to make the manuscript easier to read, also considering the comments of Reviewer 2.

The description of the methods is understandable and straightforward. I also very much like Figure 1, which nicely summarizes the methods described by the authors in subchapters 2.2 – 2.4.

Thank you! We indeed tried to summarize all the different methodologies in a clear straightforward way.

Throughout the manuscript, the authors make some hints on long-term and short-term effects of the management practices, but do not mention the age of the respective experiments and for

how long the respective practice has been applied to the individual field sites. Please add this information and include it in the interpretation of the data.

The years each experiment was running at the time of our sampling campaign, were added in Table 1 and used also in the discussion to indicate the duration of each experiment and give an indication of its effects.

In the discussion, the authors put a lot of effort in discussing the impact of management strategies on SOC in general and on the soil profile. This is of importance, for sure, but it moves the attention away from the real intention of this study, i.e. the OC fractions associated with different soil aggregates (as stated in the title). This section could be shortened with only mentioning the most important studies (see below).

Following also the recommendations of Reviewer 2, the discussion section will be shortened and focused more on the effects of the practices on the changes of organic carbon fractions among the different aggregates' sizes.

Please find some minor comments below:

Minor comments

l. 273: h-1 = ha-1:

Typo corrected

ll. 288 – 290: not necessarily what has been proposed in the Introduction and hypotheses. Please add this to the text .

Considering also the comments of reviewer 2 to reduce the length of the discussion and focus more on the important points this sentence will either be removed or rephrased to be more focused on our initial scope and hypotheses.

l. 300: I suggest to remove "with these mechanisms" – it does not become clear if mechanisms means practices or what exactly they relate to:

Expression removed as suggested.

ll. 300 – 303: this is true only for conservational managements – conventional tillage, for example, is not known to decrease mineralization as it leads to disturbances. Please re-phase this part and make clear which practices you refer to.

This part has been rephrased and shortened to follow the recommendations to focus on the core messages.

l. 311: the choice of references is not clear to me: to my knowledge, neither Blanco-Canqui & Lal (2008) nor Haddaway et al. (2017) included SOM distribution in aggregate fractions in their analyses. However, the study of Haddaway et al. (2017) is built upon a methodological framework for meta-analyses on the impacts of soil management on SOC stocks in boreo-temperate regions. This framework was then further used in the meta-analyses performed by Meurer et al. (2018). So which point do the authors want to make here? From the choice of references it is not clear if "common methodological framework" relates to the SOC measurements taken in the field, or the compilation of studies involved in analyses (for this, please see Haddaway et al. (2015) and Söderström et al. (2014)

At this point we were referring to the controversies in literature when it comes to tillage research and SOC. Indeed, the structure of the sentence is not clear. Thank you for providing us with relevant literature which can make our statement stronger. We will rephrase this section.

l. 313: I suggest to remove "alternative to inversion tillage", as it is not fully clear what the authors intent to say with that. Just leave it with "alternative practices" or different practices":

Changed to "different soil cultivation practices''

ll. 320 – 321 & 346 – 347 & 400: please mention the hypothesis – remind the reader. The same applies to l. 319 (s&c) and l. 322 (mM).

The structure will be rephrased to remind the reader to the initial hypothesis.

l. 323: please remove "especially in CZ and HU_2" or explain a bit better what is meant here.

Expression removed

ll. 327 – 332: at this point, it would be interesting to discuss if this is a methodological issue, or if the other studies simply did not further study the composition of the macroaggregates, as has been done by the authors

The fact that we did not observe differences in the macroaggregate mass does not have to do with the further study of their composition. Our initial hypothesis based on previous research was that we would expect a significant reduction of the mass as well as the carbon content of the macroaggregates caused by inversion tillage. We proceeded to further study the composition because we did not observe any important changes in the mass but only in their carbon content for the majority of the experiments. We will add a sentence in this paragraph explaining the possible reasons. We tried hard to have as uniform sampling, measuring and analysing conditions as possible across the different experiments in 5 countries. Our conclusion could possibly be linked to the time of sampling and the aggregation seasonality, which favoured the formation and preservation of large macroaggregates at expense of macroaggregates.

ll. 341 – 343: this is an important outcome and it should be further elaborated: what exactly is the benefit of large macroaggregates in relation to "good soil structure"?

The dynamics of large and small macroaggregates are of great importance for soil structure and carbon sequestration, as the formation and stability of the microaggregates which are important for long-term C storage are regulated by their quality and life cycle. A rapid turnover of macroaggregates reduces the formation of microaggregates within them and the stabilization of C within these microaggregates. Also, the soil aggregated distribution controls the presence of macro-pores that influences the flow of water, particularly near the soil surface and the existence of large macro aggregates is associated with better pore structure, optimum movement and storage of gases, water, heat and nutrients, biological activity, and exchange processes. These will be discussed in this section. Better stability during longer and intensive rainfall leads to better infiltration and less erosion and the same time to better penetration by the roots.

l. 355: here, the humification coefficient should be explained in more detail. I assume that the authors mean the amount of C that remains in the soil?

As a humification coefficient, we refer to the fraction of organic residues that are converted to more resistant soil organic matter. As we have not referred to that before and does not fall within the scope of this research, we will rephrase the sentence in order to keep the same message but without the specific terminology.

ll. 377 – 380: this is certainly true on average and depending on which studies are included in the analysis. Some studies shows that the effect of reduced tillage on SOC stocks in deeper depths might be even negative. See for example individual studies shown in Fig. 1 in Meurer et al. (2018).

Thank you for providing this interesting research. We will include both in our discussion and in the introduction where we will mention that reduced tillage may even lead to a negative change of SOC in the deeper depths.

ll. 422 – 427: this is an important point! In addition to the impracticability at field scale, potential negative environmental impacts following the high doses of organic fertilizer application should briefly be discussed here.

We will add a sentence mentioning the possible negative effects of high doses of organic fertilisers such as leaching or accumulation of nutrients with subsequent negative effects on plants, microorganisms, soil, and water.

l. 450: requires:

Typo corrected

l. 472: please be more specific with what is meant by "more time".

Our methodological design included one-time sampling and thus we cannot specify the duration of the life cycle required for a stable soil structure to be built. Here with "more time" we mean to allow the soil structure to build by not intensively disrupting through tillage.  We will try to rephrase and explain.

Literature referred to in the text:

Meurer KHE, Haddaway NR, Bolinder MA, Kätterer T. Tillage intensity affects total SOC stocks in boreo-temperate regions only in the topsoil – A systematic review using an ESM approach. Earth-Science Reviews 2018;177:613-622.

Haddaway NR, Hedlund K, Jackson LE, Kätterer T, Lugato E, Thomsen IK, Jørgensen HB, Söderström B. What are the effects of agricultural management on soil organic carbon in boreo-temperate systems? Environ Evid. 2015;4(1):1.

Söderström B, Hedlund K, Jackson LE, Kätterer T, Lugato E, Thomsen IK, Jørgensen HB. What are the effects of agricultural management on soil organic carbon (SOC) stocks? Environ Evid. 2014;3(1):1.

---

## Author Comment (AC3)

We thank the reviewer for the feedback. We improved the readability of the manuscript, shortened all the sections, and focused more on the important aspects of the research. Below we addressed all comments in detail. The comments by the reviewer are in red; our reply is in black

This manuscript (soil-2022-28) explored the effects of agricultural managements on soil aggregates and associated organic carbon fractions based on the analysis of long-term experiments in Europe. Also, authors collected a valuable dataset, including soil aggregate stability, aggregate-associated organic carbon fractions, and crop yield in 79 experimental plots under 26 treatments.

Thank you very much for appreciating the value of the dataset

Although it is meaningful research, there is not clear enough what we can expect to do on the basis of the results, especially what is the relationship between aggregate-associated organic carbon and crop yield.

Indeed, we do not focus on a possible direct relationship between the different organic carbon fractions and crop yield.

In the general literature study and discussions with the long-term experiments' owners, colleagues and stakeholders within the SoilCare project (https://www.soilcare-project.eu/), it was clear that there are no consistent differences in crop yield in function of organic carbon to be expected.

Nevertheless, we checked the crop yield for all the experiments where we sampled and found confirmation as we could not observe significant correlations with the total SOC changes in these experiments. However, we are convinced that good soil quality eventually reflects in many advantages, including a more stable crop yield by between other causes better infiltration and rooting and also less environmental negative impacts, like less erosion as an example, which we cannot prove though within our research.

Therefore, we consider information on crop yield important but not the objective. We focussed on the effects of the practices on the soil properties and did not elaborate further in this manuscript on the yield.

Very important is, however, that no significant effects on yield caused by the different practices is relevant to our research from a practical point of view. As it is mentioned in the Discussion section on the very positive side, the fact that different soil improving practices do not affect negatively in the longer term the yield stimulates their adoption from the farmers without the fear of yield loss. In the context of the SoilCare project, we often saw as a negative point an extra cost/workload to implement a new practice. Also transition requires a few years with some extra costs. Having to invest to improve the quality of the soil without yield increases requires specific support to the farmers.

My main concerns are as follows:

1.  Abstract section is written roughly without key information and data. In addition, authors should clarify the study area (countries). Are these countries representative of Europe as a whole?

    We need to limit the length of the abstract according to the guidelines of the journal but will include a list of the countries where we sampled and a short sentence on the type of data.

2.  Introduction section is too long and unfocused, especially some sentences are too long to understand (L 23-25, 43-46, and 82-84...).

    Thanks for this observation and in line with reviewer 1. The specific sentences were rephrased and shortened as well as several other sentences were shortened or split to make the manuscript easier to read, also considering the comments of Reviewer 1.

3.  Methodology section should contain a distribution map of sample points. Whether soil properties (aggregate stability and organic carbon) and crop yield are affected by latitude or climate?

The map shown below, presenting the experiments' spatial distribution has been included in the Methodology section to visualize the European gradient of the study sites as also suggested to be highlighted by Reviewer 1. Indeed, several soil properties and especially SOC are climate dependent. The decomposition of SOM is strongly controlled by temperature and soil moisture. In this research, we compare mainly the soil improving practices with the local control practice which in most cases is the business-as-usual practice and not among the different countries. We generalize only when the same trends and observations occur in each management category in all the different study sites.

[Figure]

4. Results section is too lengthy, please simplify and focus on the key points. The figures look very blurred, and please readjust your figures.

Considering all your comments and the comments of reviewer 1, the yield results section will be removed and presented in the appendix. Only one paragraph about the practical implications of not having yield differences will remain in the discussion and will refer to

the appendix for more details.  Other parts of the results section will be more focused on the significant finding to reduce the length and make the manuscript more focused.

The pictures will change according to the journal's specifications to become more readable.

5.  Discussion section is in-depth enough, but the authors seem to ignore the differences in organic carbon fractions among different sized aggregates and the relationship between aggregate-associated organic carbon and crop yield.

We will elaborate more on the first point you mention. More focus will be given in the discussion to the differences in the organic carbon fraction among the different aggregates and the interpretation of these differences. The second point as mentioned before will not be further elaborated on as that was not the focus of this research.

6.  Conclusion section should contain the limitations and prospect of your present research.

The main limitation of the research is that for logistic reasons as is mentioned in the methodology section is that we had to restrict the sampling depth to the top 15 cm. Sampling in 5 different countries and analysing the samples in our laboratory for among others water retention, carbon, aggregates and stability was a major operation. We also preferred to have enough replicates, so that statistical analysis remained feasible and consistent across the long-term experiments. The possible implications and limitations in the interpretation of the results are presented in the discussion section.  We will make it more explicit that our sampling focussed on the topsoil.

---

## Author Response (AR1)

**Response to Reviewer 1**

We would like once again to thank the Reviewer for the detailed and constructive feedback. The remarks were useful to improve the readability of the manuscript and focus more on the important aspects of the research. Below we address all comments in detail.

The comments of the reviewers are in red with replies in Black

**Comment:** The study presented by Panagea and co-authors summarizes the outcomes of the analyses of topsoil structural stability as determined by soil aggregates, and related soil organic carbon under different soil managements that are proposed beneficial for soil (structural) quality. The authors made use of a European network of long-term field experiments, which allows them to cover a wide range of region-specific management practices, as well as pedo-climatic conditions. The approach is very good and the authors collected a valuable dataset. I would think that the authors should highlight even stronger the European gradient, which is both west-east and north south.

**Reply:** Thank you very much for appreciating the value of the dataset. A map (Figure 1) as suggested by Reviewer 2 was included in the methodology section to visualize the European gradient of the research. Indeed, mentioning only the coordinates in table A.1 in the appendix is not very informative or easily accessible for the reader. The statement in the Site description section changed: "*The long-term experiments were set up independently from one another with different objectives and under different environmental conditions. Nevertheless, they offer the possibility to explore a wide range of representative management practices and pedo-climatological conditions across Europe as they cover a wide gradient both west-east and north-south (Figure 1)*".

[Figure]

*Figure 1. Study sites' location*

**Comment:** The manuscript is generally nicely written and understandable. Some sentences are long and should be split into two. For example, ll. 344 – 346 should be split into two sentences, one stating the outcomes of the study and the second one highlighting that this has also been observed in other studies. Please avoid long sentences containing too much information.

**Reply:** The specific sentence was rephrased and shortened. Several other sentences were shortened or split as well to make the manuscript easier to read, also considering the comments of Reviewer 2.

**Comment:** The description of the methods is understandable and straightforward. I also very much like Figure 1, which nicely summarizes the methods described by the authors in subchapters 2.2 – 2.4.

**Reply:** Thank you! We indeed tried to summarize all the different methodologies in a clear straightforward way.

**Comment:** Throughout the manuscript, the authors make some hints on long-term and short-term effects of the management practices, but do not mention the age of the respective experiments and for how long the respective practice has been applied to the individual field sites. Please add this information and include it in the interpretation of the data.

**Reply:** The start of the experiments was mentioned in Table A.1, but for the readers such information should be more explicitly in the main text. The years each experiment was running at the time of our sampling campaign, were added in Table 1 and used throughout the text to indicate the duration of each experiment and give an indication of its effects. Also, an indication of the scale was specified when the terms long- or short- term were used based on cited studies. E.g., line 361 in the first submission; in the revision on line 342 *"Nevertheless, the priming effect, input quality and decomposition rate seem to play a role only in the short term (~days to year as in the long-term (decades) there is evidence…"*).

**Comment:** In the discussion, the authors put a lot of effort in discussing the impact of management strategies on SOC in general and on the soil profile. This is of importance, for sure, but it moves the attention away from the real intention of this study, i.e. the OC fractions associated with different soil aggregates (as stated in the title). This section could be shortened with only mentioning the most important studies (see below).

**Reply:** Following also the recommendations of Reviewer 2, the discussion section was shortened to focus more on the effects of the practices on the changes of organic carbon fractions among the different aggregates' sizes. We believe that is important to discuss about SOC in general as this is also part of our research and is presented. Regarding the effect on the whole profile, indeed this section was detailed and shortened. A short paragraph is included though as is an important part which should not be overlooked in this kind of research and limitation in our specific analysis.

**Comment:** Please find some minor comments below:

**Minor comments** with **replies in Black**
l. 273: h-1 = ha-1:
Typo corrected

ll. 288 – 290: not necessarily what has been proposed in the Introduction and hypotheses. Please add this to the text .

Considering also the comments of reviewer 2 to reduce the length of the discussion and focus more on the important points, this sentence was removed. The whole paragraph was rephrased.

l. 300: I suggest to remove "with these mechanisms" – it does not become clear if mechanisms means practices or what exactly they relate to:

Expression removed as suggested.

ll. 300 – 303: this is true only for conservational managements – conventional tillage, for example, is not known to decrease mineralization as it leads to disturbances. Please re-phase this part and make clear which practices you refer to.

This part has been rephrased as indeed the message was wrong. We also shortened this part to follow the recommendations of reviewer 2.

l. 311: the choice of references is not clear to me: to my knowledge, neither Blanco-Canqui & Lal (2008) nor Haddaway et al. (2017) included SOM distribution in aggregate fractions in their analyses. However, the study of Haddaway et al. (2017) is built upon a methodological framework for meta-analyses on the impacts of soil management on SOC stocks in boreo-temperate regions. This framework was then further used in the meta-analyses performed by Meurer et al. (2018). So which point do the authors want to make here? From the choice of references it is not clear if "common methodological framework" relates to the SOC measurements taken in the field, or the compilation of studies involved in analyses (for this, please see Haddaway et al. (2015) and Söderström et al. (2014)

At this point, we were referring to the controversies in literature when it comes to tillage research and SOC effects resulting inconsistent methods to sample the soil, process the soil samples and analyse the effects. This may contribute to the strong controversies in the effects of the practices as is also shown in the source you provided (Meurer et al., 2018). We improved the structure of the sentence to make it clearer (line 281 in the revised version). Thank you for providing us with relevant literature which can make our statement stronger. We rephrased this section.

l. 313: I suggest to remove "alternative to inversion tillage", as it is not fully clear what the authors

intent to say with that. Just leave it with "alternative practices" or different practices":
Changed to "*different soil cultivation practices*''.

ll. 320 – 321 & 346 – 347 & 400: please mention the hypothesis – remind the reader. The same applies to l. 319 (s&c) and l. 322 (mM).

The structure was rephrased to remind the reader of the initial hypothesis as suggested.

l. 323: please remove "especially in CZ and HU_2" or explain a bit better what is meant here.

Expression removed

ll. 327 – 332: at this point, it would be interesting to discuss if this is a methodological issue, or if the other studies simply did not further study the composition of the macroaggregates, as has been done by the authors

We did not observe differences in the macroaggregate mass, so indicating that further study of their composition is important for good understanding. Our initial hypothesis based on previous research was that we would expect a significant reduction both of the mass as well as of the carbon content of the macroaggregates caused by inversion tillage. We proceeded to further study the composition because we did not observe any important changes in the mass but only in their carbon content for the majority of the experiments.

We tried hard to have as uniform sampling, measuring and analysing conditions as possible across the different experiments in 5 countries. Our conclusion could possibly be linked to the time of sampling and the aggregation seasonality, which favoured the formation and preservation of large macroaggregates at expense of macroaggregates.

We added the following sentence in this paragraph explaining the possible reasons.

*"This can be linked to the seasonal soil aggregate stability variation depending on the soil management, climatic conditions, root stage development, microbial activity and freezing-thawing processes (Dimoyiannis, 2009; Yang and Wander, 1998; Batista et al., 2022; Jirku et al., 2010), which favoured the formation and preservation of large macroaggregates at expense of macroaggregates"*

ll. 341 – 343: this is an important outcome and it should be further elaborated: what exactly is

the benefit of large macroaggregates in relation to "good soil structure"?

The dynamics of large and small macroaggregates are of great importance for soil structure and carbon sequestration, as the formation and stability of the microaggregates which are important for long-term C storage are regulated by their quality and life cycle. A rapid turnover of macroaggregates reduces the formation of microaggregates within them and the stabilization of C within these microaggregates. Also, the soil aggregated distribution controls the presence of macro-pores that influences the flow of water, particularly near the soil surface and serves as a conductive medium for light/sun radiation needed for photosynthetic cyanobacteria and algae in the topsoil. The existence of large macro aggregates is associated with better pore structure, optimum movement and storage of gases, water, heat and nutrients, biological activity, and exchange processes. Better stability during longer and intensive rainfall leads to better infiltration and less erosion and the same time to better penetration by the roots.

We added the following sentence in this paragraph explaining the benefits of large macroaggregates:

*"… which, despite being a small fraction, is important for maintaining a good soil structure. The macroaggregates presence controls the distribution of macro-pores that influence the water flow particularly near the soil surface. It is also associated with better pore connectivity, optimum movement and storage of gases, water, heat and nutrients, enhanced biological activity and root penetration."*

l. 355: here, the humification coefficient should be explained in more detail. I assume that the authors mean the amount of C that remains in the soil?

By humification coefficient, we refer to the fraction of organic materials that are converted to more resistant soil organic matter. As we have not referred to that before and since it does not fall within the scope of this research, we rephrase the sentence to keep the same message but without the specific terminology as follows: *"This may be because the exogenous organic materials (compost in BE or manure in IT_1) were at least partially decomposed and so in a more stable form (more resistant soil organic matter) (Berti et al., 2016) and of a higher quality (low C/N ratio) (Castellano et al., 2015)."*

ll. 377 – 380: this is certainly true on average and depending on which studies are included in the analysis. Some studies shows that the effect of reduced tillage on SOC stocks in deeper depths might be even negative. See for example individual studies shown in Fig. 1 in Meurer et al. (2018).

Thank you for providing this interesting research. We included this reference both in our discussion (Line 411) and in the introduction (Line 50) where we mentioned that reduced tillage may even lead to a negative change of SOC in the deeper depths.

ll. 422 – 427: this is an important point! In addition to the impracticability at field scale, potential negative environmental impacts following the high doses of organic fertilizer application should briefly be discussed here.

We added the following sentence mentioning the possible negative effects of high doses of organic fertilizers. *"Apart from the impracticability though at field scale, the application of high doses of organic fertilizers is not always preferred as it may lead to several other negative effects such as, leaching or accumulation of nutrients with subsequent negative effects on plants, soil, water, microorganisms and climate change mitigation (Song et al., 2017; Lazcano et al., 2021)."*

l. 450: requires: Typo corrected.

l. 472: please be more specific with what is meant by "more time".
Our methodological design included one-time sampling and thus we cannot specify the duration of the life cycle required for a stable aggregate to be built. Here with "more time" we mean to allow the soil structure to build by not intensively disrupting through tillage. As we cannot specify, and the expression was vague we changed this to *"i) to reduce the tillage intensity and allow for the soil structure to build and to stabilize SOM"*.

Literature referred to in the text:

Meurer KHE, Haddaway NR, Bolinder MA, Kätterer T. Tillage intensity affects total SOC stocks in boreo-temperate regions only in the topsoil – A systematic review using an ESM approach. Earth-Science Reviews 2018; 177:613-622.

**Response to Reviewer 2**

We thank once again the reviewer for the feedback. We improved the readability of the manuscript, shortened all the sections, and focused more on the important aspects of the research. Below we addressed all comments in detail.

The comments by the reviewer are in red; our reply is in black

**Comment:** This manuscript (soil-2022-28) explored the effects of agricultural managements on soil aggregates and associated organic carbon fractions based on the analysis of long-term experiments in Europe. Also, authors collected a valuable dataset, including soil aggregate stability, aggregate-associated organic carbon fractions, and crop yield in 79 experimental plots under 26 treatments.

**Reply:** Thank you very much for appreciating the value of the dataset. We felt that review papers were useful but often led to contradictory results because of differences in soil sampling and soil samples analysis methods. One important objective to us was to collect a consistent and homogeneous dataset using the same sampling and analysis methodology.

**Comment:** Although it is meaningful research, there is not clear enough what we can expect to do on the basis of the results, especially what is the relationship between aggregate-associated organic carbon and crop yield.

**Reply:** Indeed, we do not focus on a possible direct relationship between the different organic carbon fractions and crop yield. From the general literature study and discussions with the long-term experiments' owners, colleagues and stakeholders (https://www.soilcare-project.eu/), it was clear that there are no consistent differences in crop yield in function of organic carbon to be expected. Some believe very strongly, especially organic farmers, in the positive impact of organic carbon on crop yield but most refereed publications do not show a strong relationship as presented in Vonk et. al (2020).

Nevertheless, as the crop yield for all the experiments where we sampled is available, we included this information. We could not observe significant correlations with the total SOC changes in these experiments. However, we are strongly convinced that good soil quality eventually reflects indirectly many advantages, resulting in a more stable crop yield. A stable soil structure maintains better infiltration and deeper rooting by the crop, leading to better drought tolerance. Also, environmental negative impacts, like erosion due to long and intensive rains, for example, will be reduced. We did not however research these effects ourselves.

Therefore, we consider that sharing the information on crop yield is important - especially for use in future metanalysis and synthesis papers in which many times the lack of a full dataset creates major limitations (Meurer et al., 2018) - but it was not the objective of this research. We focused on the effects of the practices on the soil properties and did not elaborate further in this manuscript on the yield.

But even more important is, that no significant effects on yield were caused by the different practices. This is very relevant to our research from a practical point of view. As it is mentioned in the Discussion section (initial submission - Appendix in the current version) on the very positive side, the fact that different soil-improving practices do not affect negatively in the longer term the yield stimulates their adoption from the farmers without the fear of yield loss. In the context of the SoilCare project, we often saw as a negative point an extra cost/workload to implement a new practice. Also, the transition requires a few years with some extra costs, like investing in new tillage equipment. Having to invest to improve the quality of the soil without yield increases requires specific support to the farmers.

**Comment:** My main concerns are as follows:

**Our reply in black**

1. Abstract section is written roughly without key information and data. In addition, authors should clarify the study area (countries). Are these countries representative of Europe as a whole?

   **Reply:** We need to limit the length of the abstract according to the guidelines of the journal and keep it sharp and concise, but we included the list of the countries where we sampled and some generic data. Regarding the second part of your concern and according to the comments of Reviewer 1 to highlight even stronger the European gradient, we added in the description section the following sentence: "*The long-term experiments were set up independently from one another with different objectives and under different environmental conditions. Nevertheless, they offer the possibility to explore a wide range of representative management practices and pedo-climatological conditions across Europe as they cover a wide gradient both west-east and north-south (Figure1)*". Please note that the new Figure 1 is now the map.

2. Introduction section is too long and unfocused, especially some sentences are too long to understand (L 23-25, 43-46, and 82-84…).

   **Reply:** Thanks for this observation. The specific sentences were rephrased and shortened as well as several other sentences were shortened or split to make the manuscript easier to read, also considering the comments of Reviewer 1 who provided similar feedback.

3. Methodology section should contain a distribution map of sample points. Whether soil

properties (aggregate stability and organic carbon) and crop yield are affected by latitude or climate?

**Reply:** The map shown below, presenting the experiments' spatial distribution has been included in the Methodology section to visualize the European gradient of the study sites as suggested to be highlighted by Reviewer 1. This makes the information on the coordinates and the start of the experiments more easily accessible for the reader than providing the information in the appendix (as was the case in the submitted manuscript). The map and the length of experiments at the time of our sampling is now included in the main text. Several soil properties and especially SOC are climate dependent. The decomposition of SOM is strongly controlled by temperature and soil moisture. In this research, we investigate changes resulting from different practices, i.e we compare soil-improving practices with the local control practice which in most cases is the business-as-usual practice and not identical among the different countries. This detaches the observed effect from the climatic conditions. We only formulate generalizing conclusions when the same trends and observations occur in each management category in all the different study sites.

[Figure]

4. Results section is too lengthy, please simplify and focus on the key points. The figures look very blurred, and please readjust your figures.

**Reply:** Considering all your comments and the comments of Reviewer 1, the yield results section was removed from the main texts and is now presented in the appendix.

Other parts of the results section were removed to reduce the length and make the manuscript more focused as suggested. The pictures were also changed according to the journal's specifications.

5. Discussion section is in-depth enough, but the authors seem to ignore the differences in organic carbon fractions among different sized aggregates and the relationship between aggregate-associated organic carbon and crop yield.

   **Reply:** The differences in organic carbon fractions among different-sized aggregates are very important part of our work. All our hypotheses are formed around the differences in the mass and the OC within this and are discussed in the Discussion section. We tried to make it clearer.

The second point, connection between crop yield and aggregate-associated organic carbon, as mentioned before will not be further elaborated on as that was not the focus of this research. We strongly believe that organic carbon is very important, but the effect on yield seems to be indirect.

6. Conclusion section should contain the limitations and prospect of your present research.

The main limitation of the research is that for logistic reasons, as mentioned in the methodology section, we had to restrict the sampling depth to the top 15 cm. Sampling in 5 different countries, transporting and analyzing all the samples in our laboratory for properties like water retention, carbon, aggregates and stability. This was a major operation. We also preferred to have enough replicates, so that statistical analysis remained feasible and consistent across the long-term experiments. The possible implications and limitations in the interpretation of the results are discussed in the Discussion section. We modified though in the Conclusions the following sentence to also includes it here in short, according to your recommendations.

 *"Following testing several hypotheses and considering the limitation of the sampling depth we conclude that:….."*

Literature referred in the text:
Meurer, K. H., Haddaway, N. R., Bolinder, M. A., and Kätterer, T.: Tillage intensity affects total SOC stocks in boreo- temperate regions only in the topsoil—A systematic review using an ESM approach, Earth-Science Reviews, 177, 613–622, https://doi.org/https://doi.org/10.1016/j.earscirev.2017.12.015, 2018.

Vonk, W. J., van Ittersum, M. K., Reidsma, P., Zavattaro, L., Bechini, L., Guzmán, G., Pronk, A., Spiegel, H., Steinmann, H. H., Ruysschaert, G., and Hijbeek, R.: European survey shows poor association between soil organic matter and crop yields, Nutrient Cycling in Agroecosystems, 118, 325–334, https://doi.org/10.1007/s10705-020-10098-2, 2020